# Chat-CBM:

# Towards Interactive Concept Bottleneck Models with Frozen Large Language Models

## Abstract

Concept Bottleneck Models (CBMs) provide inherent interpretability by first predicting a set of human-understandable concepts and then mapping them to labels through a simple classifier. While users can intervene in the concept space to improve predictions, traditional CBMs typically employ a fixed linear classifier over concept scores, which restricts interventions to manual value adjustments and prevents the incorporation of new concepts or domain knowledge at test time. These limitations are particularly severe in unsupervised CBMs, where concept activations are often noisy and densely activated, making user interventions ineffective. We introduce Chat-CBM, which replaces score-based classifiers with a language-based classifier that reasons directly over concept semantics. By grounding prediction in the semantic space of concepts, Chat-CBM preserves the interpretability of CBMs while enabling richer and more intuitive interventions, such as concept correction, addition or removal of concepts, incorporation of external knowledge, and high-level reasoning guidance. Leveraging the language understanding and few-shot capabilities of frozen large language models, Chat-CBM extends the intervention interface of CBMs beyond numerical editing and remains effective even in unsupervised settings. Experiments on nine datasets demonstrate that Chat-CBM achieves higher predictive performance and substantially improves user interactivity while maintaining the concept-based interpretability of CBMs.

## 1 Introduction

With the widespread adoption of deep learning, there is a growing demand for models that are both interpretable and interactive. This need is particularly critical in domains requiring trustworthy models, such as medical applications (Klauschen et al., 2024), and in human-centered workflows requiring interactive and controllable models (Berg et al., 2019; Teso et al., 2023). Post-hoc explanation (Gunning & Aha, 2019) methods attempt to rationalize model predictions through techniques such as feature attribution (Nielsen et al., 2022) and concept-based explanations (Lee et al., 2024). However, their reliability is often questioned: potential biases in the explanation process make it difficult to separate flaws in the underlying model from artifacts of the explanation method itself (Rudin, 2019). Concept bottleneck models (CBMs) (Koh et al., 2020), in contrast, are interpretable models by design, which first map inputs to a set of human-understandable concepts and then predict class labels through this concept bottleneck (Figure 1 (a)). Crucially, the concept bottleneck also acts as an intervention interface where users can adjust concept activations to steer predictions. This user intervention ability is the key essential of CBMs and distinguishes them from alternative interpretable architectures such as the CapsuleNet (Sabour et al., 2017) and ProtoPNet (Chen et al., 2019; Xue et al., 2024).

Like other interpretable models, CBMs are subject to the well-known trade-off between interpretability and accuracy (Ras et al., 2018; Zarlenga et al., 2022). Their predictive performance often falls short of black-box counterparts, limiting adoption in domains where accuracy cannot be compromised (Sabuncu et al., 2025). To narrow this gap, recent work has explored richer concept representations, more sophisticated intervention mechanisms, and intervention-aware models (Zarlenga et al., 2022; Xu et al., 2024; Shin et al., 2023; Vandenhirtz et al., 2024; Steinmann et al., 2024; Chauhan et al., 2023). Yet, most existing CBMs still rely on score-based label predictors, which restrict user interventions to numerical edits of concept scores and prevent the addition or removal of concepts at

test time. These limitations are exacerbated in unsupervised CBMs (Oikarinen et al., 2023; Yang et al., 2023), which typically leverage CLIP-based (Radford et al., 2021) vision–language similarity over large concept banks. Lacking explicit supervision, such models often produce noisy, densely activated concept predictions (Geirhos et al., 2020; Roth et al., 2023), undermining interpretability and rendering effective user intervention nearly impossible.

In this work, we argue that these drawbacks of limited types of interventions and the ineffective intervention performance in some cases primarily stem from the reliance on score-based label predictors. We propose Chat-CBM, which shifts the inference paradigm from numerical concept activations to concept semantics by employing a language-based classifier as the CBM predictor. Chat-CBM integrates concept semantics directly into the prediction process: labels are inferred through reasoning over concept semantics rather than activation scores. This design preserves the core essentials of CBMs, the concept-based interpretability, while extending the range of possible interventions. As illustrated in Figure 1 (b), Chat-CBM enables intuitive, language-driven interventions that surpass simple score adjustments, including concept correction, addition or removal of concepts, and high-level reasoning guidance. We conduct extensive experiments

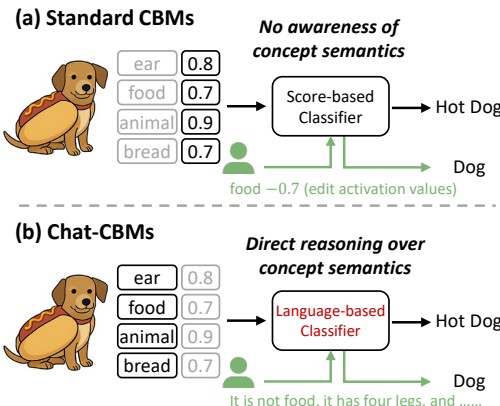

Figure 1: Illustration of standard CBMs and Chat-CBMs with score-/language-based classifiers.

across nine image classification datasets to evaluate Chat-CBM. Our results show that it outperforms traditional CBMs in classification accuracy, offers conversational interventions, and exhibits effective interventions even for unsupervised CBMs. These findings highlight the promise of language-based label predictors for building more interactive CBMs.

## 2    RELATED WORK

### 2.1    CONCEPT BOTTLENECK MODELS

**Supervised CBMs.**    For datasets with annotated concept labels, research on CBMs has primarily focused on enhancing concept representations and strengthening their intervention capabilities. For instance, CEM (Zarlenga et al., 2022) replaces scalar concept logits with learnable positive/negative embeddings; ProbCBM (Kim et al., 2023) introduces probabilistic concept embeddings to capture uncertainty; ECBM (Xu et al., 2024) employs energy-based functions over (input, concept, label) triplets to better model joint dependencies; and SCBM (Vandenhirtz et al., 2024) uses multivariate Gaussian distributions to represent correlated concept predictions. In parallel, new training paradigms and intervention policies have been explored. Interactive CBM (Chauhan et al., 2023) introduces the CooP policy, which estimates concept uncertainty to decide when user input should be requested, while IntCEM (Zarlenga et al., 2023) trains the model to actively select which concepts to query at inference. Recent works also investigate interventions, deployment under distribution shift (Zarlenga et al., 2025; He et al., 2025a), and robustness to label noise (Penaloza et al., 2025; Hu et al., 2024). Despite these advancements, supervised CBMs still rely on score-based classifiers for label prediction, which fundamentally restricts the flexibility of user interventions.

**Unsupervised CBMs.**    Obtaining fine-grained concept annotations is costly and often infeasible. To mitigate this, unsupervised CBMs have emerged, typically by constructing a concept bank using LLMs (Brown et al., 2020) or vision–language models (Bhalla et al., 2024), followed by different concept filtering strategies (Oikarinen et al., 2023; Yang et al., 2023; Yan et al., 2023; He et al., 2025b; Panousis et al., 2024; Tan et al., 2024a; Xie et al., 2025). A label predictor is subsequently trained on concept activations computed by image–text similarity. While these approaches successfully reduce annotation cost, they suffer from limited intervention capability and are difficult to integrate with advanced CBM architectures. Moreover, the large number of concepts in the bank and the dense,

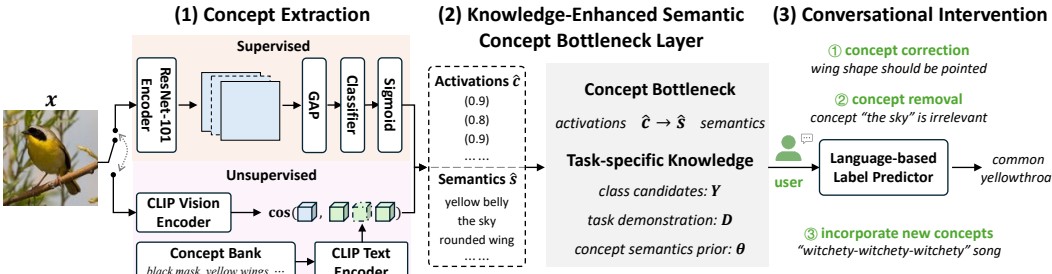

Figure 2: Overview of Chat-CBMs. We first extract concept semantics from the input images, then generate class candidates $Y$ using the baseline CBMs and incorporate the task demonstration $D$ and the concept semantics prior $\theta$ of the class candidates to form the knowledge-enhanced semantic concept bottleneck layer. Finally, the language-based label predictor $f_{\mathcal{M}}$ reasons directly in the semantic bottleneck space, producing the final predictions while supporting flexible user interventions.

noisy activations produced by CLIP features hinder the identification of actionable concepts (Roth et al., 2023), making interventions ineffective and difficult to scale.

## 2.2 CONCEPT-BASED INTERPRETABLE REASONING BEYOND LINEAR CLASSIFIERS

Beyond CBMs, several other interpretable architectures also reason over concepts. For instance, DCR constructs syntactic rule structures using concept embeddings (Barbiero et al., 2023), while CMR supports more logic-driven decision processes and enables rule-based interventions (Debot et al., 2024). Prototype-based networks learn concept prototypes (Chen et al., 2019), but the semantics of these prototypes require post-hoc analysis and, critically, do not support user interventions. XBM leverages multimodal LLMs to generate captions as concepts and then trains a BERT (Devlin et al., 2019) for downstream prediction, reducing annotation requirements but still suffering from low intervention efficiency (Yamaguchi & Nishida, 2024). Recent works have attempted to integrate concept bottleneck structures into LLMs. (Tan et al., 2024b) leverage language models with bottleneck structures to build the CBE-PLMs for interpretable text classification, but still use score-based label predictors. Similarly, CB-LLM transforms a standard LLM into a CBM framework for interpretable text classification (Sun et al., 2024), while CB-pLM follows a similar strategy but is used for protein design (Ismail et al., 2024). In contrast, our proposed Chat-CBM emphasizes the semantic information inherently available in the concept bottleneck, and importantly, explores language-based interventions that are flexible and effective for both supervised and unsupervised CBMs.

## 3 METHOD

### 3.1 PROBLEM DEFINITION AND METHOD OVERVIEW

To overcome the limitations of score-based classifiers, where concept semantics are neglected and interventions are restricted to manual edits of activation values, we propose **Chat-CBM**, an interactive CBM that employs a language-based classifier $f_{\mathcal{M}}$ for label prediction over a *knowledge-enhanced semantic concept bottleneck layer*. Unlike CBMs that operate in a numeric bottleneck of concept scores, Chat-CBM performs prediction in the semantic space of the concept bottleneck, maintaining core concept-based interpretability while extending its intervention ability.

Here we focus on the image classification task. As shown in Figure 2, Chat-CBM first obtains concept predictions $(\hat{c}, \hat{s})$ from the input image $x$, where $\hat{c} \in [0,1]^{N_c}$ denotes the activation scores of $N_c$ concepts, and $\hat{s}$ encodes their corresponding concept semantics. These are integrated into the bottleneck layer, together with task-specific knowledge. This includes class candidates $Y = \{y_i\}$ computed with a linear classifier $f(\hat{c})$, task demonstration $D$, and concept semantics prior $\theta$ for each class. A frozen LLM $f_{\mathcal{M}}$ then computes the label prediction by selecting the candidate $y_i$ with the highest probability conditioned on $D$ and $\hat{s}$:

$$P(y_i \mid D, \theta, \hat{s}) \triangleq f_{\mathcal{M}}(y_i, D, \theta, \hat{s}), \qquad \hat{y} = \underset{y_i \in Y}{\arg\max}\ P(y_i \mid D, \theta, \hat{s}). \qquad (1)$$

By explicitly situating prediction within the knowledge-enhanced semantic concept bottleneck layer, Chat-CBM preserves the interpretability-by-design property of CBMs while enabling richer and more flexible user interventions, including standard concept correction as well as flexible concept removal or integration at test time. Details of each stage are described as follows.

## 3.2 CHAT-CBM

### 3.2.1 CONCEPT EXTRACTION

As shown in Figure 2 (1), our method supports both datasets with and without concept annotations by adapting to either supervised or unsupervised CBMs for concept extraction.

**Supervised CBMs.** Given a dataset with concept labels, denoted as $\mathcal{D} = \{(\boldsymbol{x}^{(i)}, \boldsymbol{c}^{(i)}, \boldsymbol{y}^{(i)})\}$, where the $i$-th data point contains input $\boldsymbol{x}^{(i)} \in \mathcal{X}$, concept label $\boldsymbol{c}^{(i)} \in \mathcal{C} = \{0,1\}^{N_c}$, and one-hot class label $\boldsymbol{y}^{(i)} \in \mathcal{Y} = \{0,1\}^M$ of $M$ classes. As shown in Figure 2 (orange), a supervised CBM is composed of a concept predictor $g : \mathcal{X} \to \mathcal{C}$ and a class predictor $f : \mathcal{C} \to \mathcal{Y}$. To mitigate the possible concept leakage problem and improve intervention efficiency (Havasi et al., 2022), we train the concept predictor $g(\cdot)$ and label predictor $f(\cdot)$ independently. The concept activation values $\hat{\boldsymbol{c}}$ and semantics $\hat{\boldsymbol{s}}$ are obtained by:

$$\hat{\boldsymbol{c}} = g(\boldsymbol{x}), \qquad \hat{\boldsymbol{s}} = \text{decode}(\hat{\boldsymbol{c}}), \tag{2}$$

where $\text{decode}(\cdot)$ returns the concept semantics when activation values are larger than 0.5.

**Unsupervised CBMs.** For datasets $\mathcal{D} = \{(\boldsymbol{x}^{(i)}, \boldsymbol{y}^{(i)})\}$ without concept annotations, we build unsupervised CBMs for concept extraction. As shown in Figure 2, we first adopt the concept bank $\mathcal{B} = \{t_1, \cdots, t_{N_c}\}$ from (Yang et al., 2023; He et al., 2025b), which consists of $N_c$ concepts. Then, we leverage CLIP to encode visual features of the input image and textual features of each concept in the concept bank, and compute the cosine similarity as the concept activation values $\hat{\boldsymbol{c}}$:

$$\hat{\boldsymbol{c}} = g(\boldsymbol{x}) = \cos(e_v(\boldsymbol{x}), e_t(\mathcal{B})) \in \mathbb{R}^{N_c}, \tag{3}$$

where $e_v$ and $e_t$ are CLIP vision and text encoders, respectively. Then the semantics of the top-10 activated concepts are used by the label predictor. (The top-N choice is discussed in appendix A.3.)

### 3.2.2 KNOWLEDGE-ENHANCED SEMANTIC CONCEPT BOTTLENECK LAYER

As illustrated in Figure 2 (2), we then construct the knowledge-enhanced semantic concept bottleneck, including the concept semantics $\hat{\boldsymbol{s}}$, the class candidates $\boldsymbol{Y}$, a demonstration set $\boldsymbol{D}$ consists of in-context learning (ICL) examples, and integrate the concept semantics prior $\boldsymbol{\theta}$ for the candidate classes.

**Class Candidates Generation.** In order to obtain the class candidates $\boldsymbol{Y}$ for Chat-CBM, we use the label predictor $f(\cdot)$ of the baseline CBM and take the top-N predictions as $\boldsymbol{Y}$:

$$\boldsymbol{Y} = \{\boldsymbol{y}_1, \cdots, \boldsymbol{y}_N\} = \text{top-N}(f(\hat{\boldsymbol{c}})). \tag{4}$$

**In-Context Learning Examples Selection.** To enhance the reasoning capability of Chat-CBMs with frozen LLMs, we employ ICL to encourage LLMs to learn the associations between concepts and labels in the demonstration and, accordingly, make the right prediction. For each answer candidate $\boldsymbol{y}_i$ in $Y$, we randomly select $K$ samples of class $\boldsymbol{y}_i$ from the val set with their predicted concept semantics $\hat{\boldsymbol{s}}_{\text{val}}$ to form the ICL demonstrations:

$$\boldsymbol{D} = \{I, (\hat{\boldsymbol{s}}_{\text{val}}^{(1)}, \boldsymbol{y}_1), \cdots, (\hat{\boldsymbol{s}}_{\text{val}}^{(K)}, \boldsymbol{y}_1), \cdots, (\hat{\boldsymbol{s}}_{\text{val}}^{(K)}, \boldsymbol{y}_N)\}, \tag{5}$$

where $I$ is the task instruction with format control like "*Answer the image class based on the concepts, the answer format is <analysis: ...,> <answer: ...>*".

**Class Concept Semantics Prior Integration.** Beyond enhancing the local mapping relationships between concepts and class labels via ICL, another advantage of using LLMs is that they also allow the incorporation of global task-specific knowledge. To further enrich reasoning, we optionally

**Input Image**    **Concept Semantics**    **Task-specific Knowledge**

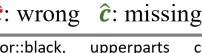

$\hat{c}$: wrong   $\hat{c}$: missing

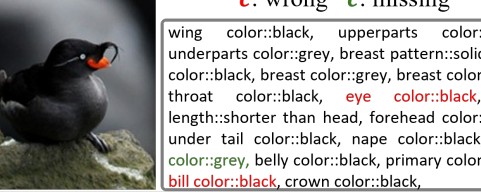

wing color::black, upperparts color::black, underparts color::grey, breast pattern::solid, back color::black, breast color::grey, breast color::black, throat color::black, eye color::black,, bill length::shorter than head, forehead color::black, under tail color::black, nape color::black, belly color::grey, belly color::black, primary color::black, bill color::black, crown color::black,

**Class Candidates:**
$f(\hat{c})$ = rhinoceros auklet, crested auklet, …
**Task Demonstration:**
<val set predicted concepts> + <class candidates> + <answer>, the answer format is <analysis: …>, <answer: …>
**Concept Semantics Prior:**
1) Rhinoceros Auklet is usually ….
2) Crested Auklet is usually ….

**Analysis:** Based on the given concepts and the demonstrations, rhinoceros auklet and crested auklet share many features such as primary color is black, wing color is black and throat color is black. Given these observations and the fact that the other features match closely with both species, but the presence of **black eye color** and **black bill color** (**wrong concepts**), tilts the balance towards a rhinoceros auklet.
**Prediction:** rhinoceros auklet (wrong)
================================**User Intervention Case 1** ================================
**Conversational Intervention:**
user input: the eye color is white, and the bill is orange-red. (**standard concept correction**)
**Analysis:**
Then consider the shared features and the presence of orange-red bill, which is the key features of crested auklet, this leads to crested auklet.
**Prediction:** crested auklet (right)
================================**User Intervention Case 2** ================================
**Conversational Intervention:**
user input: There is a forward-arching feather on the head. (**concepts augmentation**)
**Analysis:**
Crested auklet is known for its colorful plumage with a forehead crest, if the bird has a forward-arching feather on the head, it strongly indicates the Crested Auklet.
**Prediction:** crested auklet (right)
================================**User Intervention Case 3** ================================
**Conversational Intervention:**
user input: do not answer based solely on one or two isolated concepts, please compare the whole concept set to the demonstrations. (**reasoning strategy guidance**)
**Analysis:**
Let's carefully compare the concepts provided with the demonstrations The given concept set is closer to the last two examples, which are classified as crested auklet. The given concepts do not include the specific concepts like shape is duck-like that are associated with rhinoceros auklet in the other demonstrations. Based on this comparison, the most similar concept set is from the crested auklet examples.
**Prediction:** crested auklet (right)

Figure 3: Conversational intervention on the CUB dataset. Green highlights the positive reasoning and blue highlights the negative reasoning process. Users can either directly correct concept predictions like standard CBMs (**case 1**), adding new (or removing) concepts beyond the predefined concept bottleneck (**case 2**), or give a high-level reasoning strategy to guide thinking (**case 3**).

augment the input with structured class prior knowledge $\boldsymbol{\theta}$, which describes the most common attributes for each candidate class $\boldsymbol{y}_i$. The inference objective then becomes:

$$\hat{\boldsymbol{y}} = \arg\max_{\boldsymbol{y}_i \in \boldsymbol{Y}} P(\boldsymbol{y}_i \mid \boldsymbol{D}, \boldsymbol{\theta}, \hat{\boldsymbol{s}}), \qquad (6)$$

where $\boldsymbol{\theta}$ serves as an additional global class prior and can take various forms. Here we simply use the average concept of the class from the training set as $\boldsymbol{\theta}$, with details in appendix A.5.

### 3.2.3 INTERVENTION

**Standard Numerical Intervention.** Chat-CBMs retain the standard intervention abilities of CBMs, allowing users to directly edit concept activation values ($u(\cdot)$ denotes user intervention). Given updated activations $\hat{\boldsymbol{c}}_{\text{new}} = u(\hat{\boldsymbol{c}})$, the corresponding concept semantics $\hat{\boldsymbol{s}}_{\text{new}}$ and class candidates $\boldsymbol{Y}_{\text{new}}$ are passed to the language-based classifier for inference.

**Conversational Intervention.** Beyond numerical edits, the language-based classifier enables flexible interventions via natural language $u_{\text{text}}$. We highlight three representative types of conversational interventions and provide examples in Figure 3:

Table 1: Classification accuracy on datasets with concept labels. We report the mean and standard deviation from five runs with different random seeds. (LLaMA-3-70B-Instruct for Chat-CBM.)

| Data / Model | CUB | | AwA2 | | PBC | |
|---|---|---|---|---|---|---|
| Metric | Concept Acc. | Class Acc. | Concept Acc. | Class Acc. | Concept Acc. | Class Acc. |
| End-to-End | - | $0.825 \pm 0.002$ | - | $0.953 \pm 0.001$ | - | $0.997 \pm 0.000$ |
| Hard CBM | $0.960 \pm 0.004$ | $0.708 \pm 0.003$ | $0.980 \pm 0.001$ | $0.901 \pm 0.001$ | $0.920 \pm 0.005$ | $0.959 \pm 0.011$ |
| ProbCBM | $0.955 \pm 0.003$ | $0.723 \pm 0.001$ | $0.959 \pm 0.000$ | $0.890 \pm 0.007$ | $0.950 \pm 0.001$ | $0.990 \pm 0.002$ |
| CEM | $0.962 \pm 0.002$ | $0.799 \pm 0.003$ | $0.979 \pm 0.003$ | $0.924 \pm 0.002$ | $0.952 \pm 0.002$ | $0.993 \pm 0.001$ |
| SCBM | $0.967 \pm 0.002$ | $0.778 \pm 0.004$ | $0.983 \pm 0.000$ | $0.917 \pm 0.001$ | $0.956 \pm 0.001$ | $0.992 \pm 0.002$ |
| CBM-MLP(1) | $0.965 \pm 0.009$ | $0.791 \pm 0.001$ | $0.982 \pm 0.000$ | $0.945 \pm 0.000$ | $0.956 \pm 0.003$ | $\mathbf{0.995} \pm 0.001$ |
| CBM-MLP(2) | $0.965 \pm 0.009$ | $0.768 \pm 0.002$ | $0.982 \pm 0.000$ | $0.943 \pm 0.000$ | $0.956 \pm 0.003$ | $\mathbf{0.995} \pm 0.001$ |
| CBM | $0.965 \pm 0.009$ | $0.752 \pm 0.005$ | $0.982 \pm 0.000$ | $0.923 \pm 0.004$ | $0.956 \pm 0.003$ | $0.988 \pm 0.008$ |
| + Chat-CBM | $0.965 \pm 0.009$ | $0.815 \pm 0.005$ | $0.982 \pm 0.000$ | $\mathbf{0.964} \pm 0.002$ | $0.956 \pm 0.003$ | $0.986 \pm 0.002$ |
| ECBM | $0.967 \pm 0.003$ | $0.806 \pm 0.004$ | $0.983 \pm 0.001$ | $0.916 \pm 0.000$ | $0.935 \pm 0.004$ | $0.994 \pm 0.001$ |
| + Chat-CBM | $0.967 \pm 0.003$ | $\mathbf{0.816} \pm 0.006$ | $0.983 \pm 0.001$ | $0.961 \pm 0.005$ | $0.935 \pm 0.004$ | $0.989 \pm 0.001$ |

Table 2: Classification accuracy on datasets without concept labels. We report the mean and standard deviation from five runs with different random seeds. (Qwen2.5-32B-Instruct for Chat-CBM)

| Data / Model | DTD | Food-101 | Flower-102 | CIFAR10 | CIFAR100 | ImageNet |
|---|---|---|---|---|---|---|
| Linear Prob (All) | $0.821 \pm 0.003$ | $0.952 \pm 0.000$ | $0.993 \pm 0.001$ | $0.981 \pm 0.000$ | $0.873 \pm 0.001$ | $0.841 \pm 0.003$ |
| Linear Prob (1-shot) | $0.436 \pm 0.010$ | $0.578 \pm 0.004$ | $0.477 \pm 0.003$ | $0.624 \pm 0.003$ | $0.393 \pm 0.008$ | $0.422 \pm 0.004$ |
| Linear Prob (2-shot) | $0.537 \pm 0.001$ | $0.749 \pm 0.001$ | $0.610 \pm 0.003$ | $0.803 \pm 0.002$ | $0.574 \pm 0.003$ | $0.558 \pm 0.005$ |
| LaBo (All) | $0.769 \pm 0.001$ | $0.924 \pm 0.005$ | $0.993 \pm 0.001$ | $0.978 \pm 0.001$ | $0.860 \pm 0.002$ | $0.840 \pm 0.006$ |
| LaBo (1-shot) | $0.531 \pm 0.016$ | $0.806 \pm 0.009$ | $0.825 \pm 0.003$ | $0.910 \pm 0.002$ | $0.627 \pm 0.007$ | $0.512 \pm 0.014$ |
| LaBo (2-shot) | $0.552 \pm 0.004$ | $0.840 \pm 0.002$ | $0.895 \pm 0.001$ | $0.910 \pm 0.001$ | $0.658 \pm 0.003$ | $0.571 \pm 0.008$ |
| **LaBo-Chat-CBM** (2-shot) | $0.677 \pm 0.011$ | $0.753 \pm 0.003$ | $0.876 \pm 0.002$ | $0.889 \pm 0.002$ | $0.670 \pm 0.004$ | $0.601 \pm 0.002$ |
| V2C-CBM (All) | $0.782 \pm 0.003$ | $0.927 \pm 0.002$ | $0.987 \pm 0.002$ | $0.980 \pm 0.000$ | $0.864 \pm 0.000$ | $0.841 \pm 0.002$ |
| V2C-CBM (1-shot) | $0.421 \pm 0.017$ | $0.586 \pm 0.024$ | $0.884 \pm 0.009$ | $0.893 \pm 0.008$ | $0.627 \pm 0.015$ | $0.561 \pm 0.009$ |
| V2C-CBM (2-shot) | $0.492 \pm 0.003$ | $0.745 \pm 0.005$ | $0.930 \pm 0.009$ | $0.934 \pm 0.002$ | $0.651 \pm 0.003$ | $0.615 \pm 0.005$ |
| **V2C-Chat-CBM** (2-shot) | $0.734 \pm 0.004$ | $0.786 \pm 0.019$ | $0.914 \pm 0.002$ | $0.955 \pm 0.007$ | $0.727 \pm 0.002$ | $0.667 \pm 0.004$ |

- **Concept correction**: Standard corrections can also be performed conversationally with awareness of prior reasoning, e.g., "*the concept forest is wrongly predicted*"(Figure 3 case 1).
- **Concept augmentation/removal**: New concepts $s_{\text{new}} \notin \hat{s}$ can be added, or existing ones $s_i \in \hat{s}$ removed, via prompts such as "*the bird also has a forward-arching feather on the head*" or "*ignore concepts about bird size during analysis*"(Figure 3 case 2).
- **High-level strategy guidance**: Users can provide high-level reasoning strategies, e.g., "*focus on the bird size when distinguishing common yellowthroat and yellow-breasted chat*"(Figure 3 case 3).

Formally, intervention messages $u_{\text{text}}$ are incorporated into the conversation history $\mathcal{H}$, and new predictions are generated as:

$$\hat{y}_{\text{new}} = \arg \max_{y_i \in Y} P(y_i \mid D, \theta, \hat{s}, \mathcal{H}, u_{\text{text}}). \tag{7}$$

This process allows users to understand and take control of the decision pipeline.

## 4 EXPERIMENTS

**Datasets.** We employed two types of datasets to validate the effectiveness of our approach in both supervised and unsupervised CBMs. Datasets with concept labels: (1) CUB (Wah et al., 2011), a fine-grained bird classification dataset, we follow (Koh et al., 2020) to use 112 concepts, (2) AwA2 (Xian et al., 2018), which contains 50 animal classes with 85 attributes, and (3) PBC (Acevedo et al., 2020), a white blood cell classification dataset with 5 white blood cell classes and 11 morphological attributes (31 concepts) from (Tsutsui et al., 2023). Datasets without concept labels: (1) DTD (Cimpoi et al., 2014) for abstract texture classification of 47 classes, (2) Food-101 (Bossard et al., 2014) with 101 types of food, (3) Flower-102 (Nilsback & Zisserman, 2008) for fine-grained classification of 102 types of flowers, and (4) CIFAR10, (5) CIFAR100 (Krizhevsky & Hinton, 2009), (6) ImageNet (Russakovsky et al., 2015) as standard classification benchmarks. For all datasets, we use the same data split settings for training and evaluating the performance of different methods.

**Implementation Details.** We compare our Chat-CBM to both **(1)** supervised CBMs, including CBM (Koh et al., 2020), Hard CBM which uses 0/1 activation values for CBM (Havasi et al., 2022), ProbCBM (Kim et al., 2023), CEM (Zarlenga et al., 2022), SCBM (Vandenhirtz et al., 2024), ECBM (Xu et al., 2024) and also CBM with non-linear heads: CBM-MLP(1), CBM-MLP(2) for 1-layer and 2-layer MLP. **(2)** unsupervised CBMs such as LaBo (Yang et al., 2023) and V2C-CBM (He et al., 2025b). We use ResNet-101 (He et al., 2016) as the backbone for supervised CBMs and CLIP ViT-L/14 (Radford et al., 2021) for unsupervised CBMs. And we test Chat-CBMs with LLaMA3-Instruct (Dubey et al., 2024) and Qwen2.5-Instruct (Yang et al., 2024) as the language-based classifiers. We use AdamW (Loshchilov & Hutter, 2019) and ConsineAnnealingLR for training all baseline models. All images are resized to 224×224 for both training and testing. We use the same training settings for each dataset and report the mean and standard deviation across five runs with different random seeds. Full details are provided in appendix A, and we also provide additional experiments in appendix B and more visualization results with failure case analysis in appendix C.

## 4.1 CLASSIFICATION PERFORMANCE

**Compared to Supervised CBMs.** Table 1 presents the classification accuracy of different methods on datasets with concept labels. Chat-CBM with frozen LLaMA3-70B-Instruct surpasses the baselines on the CUB and AwA2 datasets, and Table 5 also exhibits results with different LLM sizes and series. For the PBC dataset, we found that standard CBMs (Koh et al., 2020), although exhibit high classification accuracy, suffer from severe concept leakage problems (Havasi et al., 2022), because the intervention procedure is ineffective for them as shown in Figure 4, but our Chat-CBM can achieve better performance compared with Hard CBMs (Havasi et al., 2022) and also show an effective intervention curve. We also test our Chat-CBMs with ECBMs (Xu et al., 2024) as baselines, and Chat-CBMs achieve better classification performance due to the improvement of the baseline.

**Compared to Unsupervised CBMs.** For datasets without concept labels, the results are shown in Table 2. Chat-CBM can achieve comparable performance under few-shot configurations, but is inferior to LaBo and V2C-CBM under the all-shot setting, though it's equipped with numerous parameters. This gap is largely due to the noisy concept bottlenecks in VL-CBMs, while Chat-CBM leverages only the top-activated concepts and frozen LLMs without any fine-tuning. Interestingly, Chat-CBMs with V2C-CBMs as baselines demonstrate better performance than LaBo baselines, and we think this is because the concept bank of V2C-CBMs contains more accurate and concise visual concepts compared to LaBo, as discussed in (He et al., 2025b). It may need additional effort to design more appropriate concept extraction methods for achieving competitive performance on unsupervised datasets with language-based label predictors, but currently, we choose to focus on exploring the intervention capabilities of Chat-CBM on unsupervised datasets.

## 4.2 INTERVENTION

Here we experiment with different types of intervention methods to validate the interactivity and intervention capabilities of Chat-CBMs, including **(A)** standard numerical intervention, **(B)** concept correction or removal, and **(C)** concept augmentation. We also evaluate the capability of intervention via high-level reasoning strategy guidance in appendix B.1 and conduct a small user study to validate the interactivity of Chat-CBMs to end-users in appendix B.2.

**(A) Concept Correction via Standard Numerical Intervention.** We begin by evaluating standard concept correction on datasets with annotated concepts, following (Koh et al., 2020; Xu et al., 2024) and using a random policy where we fix the intervention ratio and randomly select unintervented concepts for a given sample. Specifically, we intervene on the baseline CBMs and update $\hat{s}$ and $Y$ accordingly for Chat-CBM. The results in Figure 4 highlight the effectiveness of Chat-CBM interventions, particularly on the CUB dataset, where fine-grained distinctions rely heavily on specific concepts. On the PBC dataset, independent CBMs suffer from severe concept leakage, which undermines intervention effectiveness. In contrast, Chat-CBM reasons directly in the semantic space of concepts through its language-based classifier, rather than relying on raw activation values, which prevents the label predictor from exploiting spurious class proxies and thereby yields consistent performance gains under intervention.

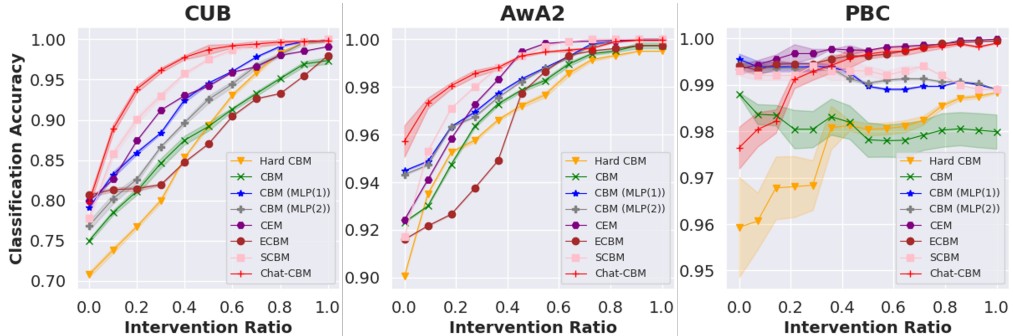

Figure 4: Intervention via Standard Numerical Intervention. (Chat-CBM with LLaMA3-8B-Instruct)

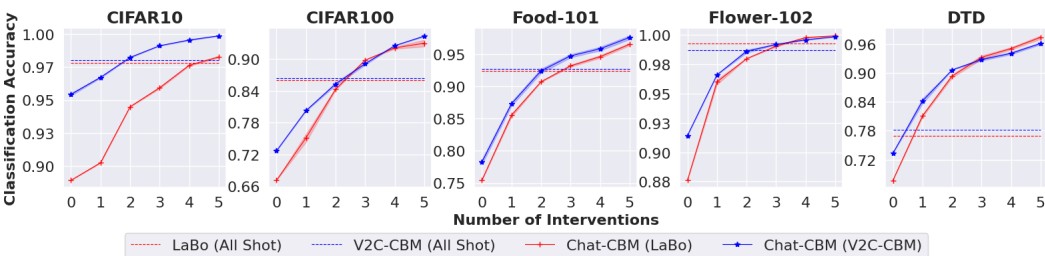

Figure 5: Intervention via Simulated Concept Correction or Removal. The LLM for Chat-CBM is Qwen2.5-32B-Instruct with the same setting in Table 2. We use an assistant MLLM (Qwen2.5-VL-7B) to simulate user intervention (where only the input image and Chat-CBM's predictions are available), and the MLLM only emphasizes or removes one concept at each step. The all-shot performance of baseline models is provided for reference.

**(B) Intervention via Simulated Concept Correction or Removal.** Direct user interventions on existing unsupervised CBMs are nearly infeasible. These models typically rely on hundreds or even thousands of concepts for reasoning, making it impractical to identify actionable concepts. To automatically conduct interventions for Chat-CBMs, we employ an assistant multimodal large language model (MLLM) to simulate user interventions. We provide the input image, the top-20 predicted concepts, and the conversation history of Chat-CBM to the MLLM (Qwen2.5-VL-7B), then prompt the MLLM to correct (emphasize/remove) **only one** concept, and we prompt the MLLM again if the candidate class labels exist in its output. We use the top-10 label predictions as class candidates, ensuring a high accuracy upper bound. As shown in Figure 5, Chat-CBM demonstrates effective performance gains for unsupervised CBMs under these simulated user interventions. Details are provided in appendix A.7.

**(C) Intervention via Concept Augmentation.**
We further test the situation when new concepts are introduced during test time. We first design a controlled setting where we train CBMs on the CUB and AwA2 datasets with incomplete concepts, and the rest are used as new concepts for intervention. The results are shown in Figure 6. Because some concepts are useful for classification, and may be easily recognized by users (such as the *forward-arching feather on the head* in Figure 3 case 2), or can come from beyond the images (such as sounds and smell). We also explore the descriptions of the target class from Wikipedia and replace the class names with general names like "the bird" or "the animal", and then

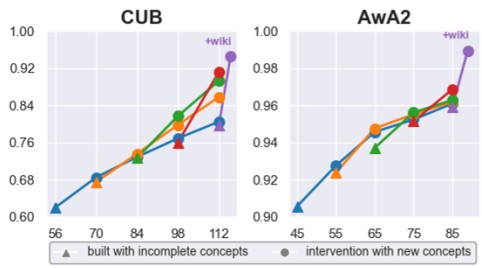

Figure 6: Intervention via Concept Augmentation (Chat-CBM with LLaMA3-8B-Instruct).

provide them to Chat-CBMs as new information at test-time. Although the huge performance improvement, as shown in Figure 6 (+wiki), seems obvious

Table 3: Ablation on knowledge enhancement strategies for the semantic concept bottleneck layer.

| Model | Strategy | | | Supervised Dataset | | | Unsupervised Dataset | |
|---|---|---|---|---|---|---|---|---|
| | $D$ | $D$-GT | $\theta$ | CUB | AwA2 | PBC | Flower-102 | DTD |
| Chat-CBM (Qwen2.5-7B) | ✔ | | | $0.645 \pm 0.007$ | $0.722 \pm 0.005$ | $0.810 \pm 0.007$ | $0.853 \pm 0.007$ | $0.665 \pm 0.022$ |
| | ✔ | ✔ | | $0.655 \pm 0.011$ | $0.754 \pm 0.006$ | $0.830 \pm 0.009$ | $0.843 \pm 0.021$ | $0.629 \pm 0.017$ |
| | ✔ | | ✔ | $0.771 \pm 0.008$ | $0.817 \pm 0.003$ | $0.868 \pm 0.003$ | $0.862 \pm 0.009$ | $0.675 \pm 0.009$ |
| | ✔ | ✔ | ✔ | $\mathbf{0.775} \pm 0.013$ | $\mathbf{0.871} \pm 0.010$ | $\mathbf{0.878} \pm 0.007$ | $\mathbf{0.899} \pm 0.012$ | $\mathbf{0.710} \pm 0.011$ |
| Chat-CBM (Qwen2.5-14B) | ✔ | | | $0.734 \pm 0.006$ | $0.930 \pm 0.019$ | $0.899 \pm 0.002$ | $0.897 \pm 0.005$ | $0.689 \pm 0.007$ |
| | ✔ | ✔ | | $0.738 \pm 0.011$ | $0.932 \pm 0.004$ | $0.927 \pm 0.004$ | $0.848 \pm 0.011$ | $0.685 \pm 0.015$ |
| | ✔ | | ✔ | $0.776 \pm 0.022$ | $0.945 \pm 0.004$ | $0.949 \pm 0.002$ | $0.902 \pm 0.003$ | $0.701 \pm 0.008$ |
| | ✔ | ✔ | ✔ | $\mathbf{0.801} \pm 0.003$ | $\mathbf{0.951} \pm 0.002$ | $\mathbf{0.965} \pm 0.003$ | $\mathbf{0.910} \pm 0.005$ | $\mathbf{0.722} \pm 0.013$ |

Table 4: Ablation on the number of ICL examples.

| Data | CUB | | | AwA2 | | | WBC | | | Flower-102 | | |
|---|---|---|---|---|---|---|---|---|---|---|---|---|
| Model | N2-K1 | N2-K2 | N2-K3 | N2-K1 | N2-K2 | N2-K3 | N5-K1 | N5-K3 | N5-K5 | N2-K1 | N2-K2 | N2-K3 |
| Chat-CBM (LLaMA3-8B) | 0.784 | 0.797 | 0.798 | 0.910 | 0.957 | 0.965 | 0.930 | 0.976 | **0.980** | 0.893 | 0.915 | 0.914 |
| Chat-CBM (Qwen2.5-7B) | 0.735 | 0.775 | 0.782 | 0.871 | 0.871 | 0.898 | 0.827 | 0.878 | 0.913 | 0.856 | 0.899 | 0.912 |
| Chat-CBM (Qwen2.5-14B) | 0.789 | 0.801 | **0.802** | 0.936 | 0.951 | **0.962** | 0.945 | 0.965 | 0.972 | 0.901 | 0.910 | **0.916** |

Table 5: Ablation on different LLMs and LLM sizes for Chat-CBMs.

| | LLM | LLaMA3-Instruct | | Qwen2.5-Instruct | | | | |
|---|---|---|---|---|---|---|---|---|
| Data | Baseline | 8B | 70B | 0.5B | 7B | 14B | 32B | 72B |
| CUB | | $0.797 \pm 0.006$ | $\mathbf{0.815} \pm 0.005$ | $0.725 \pm 0.009$ | $0.775 \pm 0.013$ | $0.801 \pm 0.003$ | $0.803 \pm 0.004$ | $\underline{0.812} \pm 0.002$ |
| AwA2 | CBM | $0.957 \pm 0.007$ | $\mathbf{0.964} \pm 0.002$ | $0.782 \pm 0.009$ | $0.871 \pm 0.010$ | $\underline{0.951} \pm 0.002$ | $0.949 \pm 0.002$ | $0.950 \pm 0.001$ |
| PBC | | $0.976 \pm 0.010$ | $\mathbf{0.986} \pm 0.002$ | $0.862 \pm 0.010$ | $0.878 \pm 0.007$ | $0.965 \pm 0.002$ | $0.975 \pm 0.001$ | $\underline{0.976} \pm 0.001$ |
| CIFAR10 | | $0.929 \pm 0.007$ | $\underline{0.951} \pm 0.006$ | $0.810 \pm 0.023$ | $0.950 \pm 0.005$ | $0.951 \pm 0.012$ | $0.955 \pm 0.007$ | $\mathbf{0.956} \pm 0.005$ |
| Flower-102 | V2C-CBM | $0.915 \pm 0.008$ | $\mathbf{0.933} \pm 0.002$ | $0.822 \pm 0.017$ | $0.899 \pm 0.012$ | $0.910 \pm 0.005$ | $0.914 \pm 0.002$ | $\underline{0.921} \pm 0.003$ |
| DTD | | $0.731 \pm 0.013$ | $\mathbf{0.757} \pm 0.009$ | $0.606 \pm 0.029$ | $0.710 \pm 0.011$ | $0.722 \pm 0.013$ | $\underline{0.734} \pm 0.004$ | $0.732 \pm 0.009$ |

because of the rich information from Wikipedia, we want to argue that previous activation-based classifiers don't easily support receiving such kind of new information at test-time, and this remains a distinctive advantage of language-based CBMs (detailed implementations in appendix A.8).

## 4.3 ABLATION STUDY

**Ablation on the Knowledge-Enhanced Semantic Concept Bottleneck.** We further conduct an ablation study on diverse input configurations to validate the efficacy of our knowledge injection strategies for our semantic concept bottleneck, with quantitative results presented in Table 3. Strategy $D$ implies augmenting the ICL prompt with the baseline's class candidates $Y_{\text{val}}$, whereas $D$-GT (our default setting) uses the standard format containing only concepts and labels. Strategy $\theta$ means integrating dataset-specific class prior knowledge $\theta$, with detailed implementations in appendix A.5. Our analysis reveals that incorporating prior knowledge yields significant improvements in Chat-CBM's classification accuracy. This enhancement stems from a key observation: while the concept predictor achieves reasonably high accuracy at the concept level, the overall concept accuracy remains suboptimal (as noted in Xu et al. (2024)). Individual concept prediction errors can consequently misguide Chat-CBM's final decisions. The introduced prior knowledge effectively enhances Chat-CBM's robustness against noise in concept predictions (e.g., adding $\theta$ improves the accuracy on the CUB dataset from 0.645 to 0.771 as shown in Table 3). We also examine how the number of ICL examples affects Chat-CBM's performance (Table 4). Chat-CBM demonstrates strong few-shot learning capabilities inherent to modern LLMs, with classification performance scaling positively with the number of ICL examples. However, we note that increasing ICL examples linearly expands context length, and further scaling ICL requires additional computational resources and larger LLM architectures to better leverage the information.

**Ablation on Different LLMs and LLM Sizes.** We further evaluate Chat-CBM with different LLM backbones and model sizes, as shown in Table 5. Larger LLMs generally achieve better classification performance than their small counterparts. This can be attributed to their stronger ability to capture the in-context mappings and to leverage the provided class prior knowledge, which is crucial when the concept inputs are noisy. On the Qwen2.5-Instruct series, we observe that increasing the model size beyond 14B does not lead to further significant improvements. This suggests that the model capacity is no longer a limiting factor—i.e., 14B is already sufficient to encode the necessary task

structure and priors. Combined with the prompt ablation results in Table 3, we hypothesize that ICL combined with class priors provides adequate supervision, and further gains rely more on the model's robustness to noisy or imperfect concept inputs than on increased parameter count. For practical applications, we see two viable paths: (1) using larger frozen LLMs via APIs (cloud deployment) for maximum reasoning capability, or (2) fine-tuning compact models (like the 0.5B variant) on task-specific data for edge deployment. Our current results with *frozen* small models serve as a lower-bound baseline.

## 5 CONCLUSION AND LIMITATIONS

We introduce Chat-CBM, which replaces the score-based classifier of conventional CBMs with a language-based predictor operating in a semantic concept bottleneck. This design preserves the concept-based interpretability by explicitly keeping a concept bottleneck structure, while extending the intervention ability beyond numeric edits. Experiments on both annotated and unannotated datasets show that Chat-CBM improves classification accuracy, supports multiple forms of intervention, and scales with ICL examples and model size. Limitations remain: the use of semantic concept bottlenecks prevents the concept leakage problem, but also limits the representation ability under an incomplete concept situation. In addition, Chat-CBMs have no awareness of concept uncertainty or absence, which may omit useful information. The use of LLMs introduces extra inference cost, and deployment in sensitive domains requires safeguarding against harmful knowledge encoded in LLMs. More detailed discussions are provided in Appendix D.

**Ethics Statement** This work uses only publicly available datasets (CUB, AwA2, PBC, DTD, Food-101, Flower-102, CIFAR, and ImageNet). The PBC dataset consists of anonymized white blood cell images without patient-identifiable information, and thus does not raise additional privacy concerns. No human subjects or personally identifiable data were involved in this study. Chat-CBM leverages frozen large language models as label predictors. While LLMs may encode undesirable biases or potentially harmful knowledge, our experiments are restricted to benchmark datasets and do not involve deployment in sensitive domains. Future applications in areas such as healthcare or law will require additional safeguards to ensure fairness, reliability, and safe use.

**Reproducibility statement** We state that Chat-CBM is highly reproducible. Appendix A encompasses the hyperparameters, computational resources, and implementation details of our method and experiments. Section 3.2 explains how Chat-CBM works and also provides example cases. Our method, Chat-CBM, can be easily implemented with existing CBM architectures, which provide concept prediction results. The use of publicly available parameter frozen LLMs also enhances the reproducibility of Chat-CBMs.

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

# A    ALL IMPLEMENTATION DETAILS

## A.1    TRAINING DETAILS

For all datasets with concept labels, we use a ResNet-101 pretrained on ImageNet1k as the concept predictor backbone for all models, including CBM, CBM-MLP(1), CBM-MLP(2), ProbCBM, CEM, ECBM, and SCBM, and use a single layer as the label predictor except for those architectures with advanced designs (such as ProbCBM and ECBM). The hidden dimension of the MLP is 256, 128, 64 for CUB, AwA2, and PBC datasets separately, and we use ReLU as non-linearity for the MLP. We use AdamW as the optimizer and CosineAnnealingLR as the learning rate scheduler for training all models. All images are resized to 224×224 for both training and testing. For all baseline models on datasets with concept labels, we train the concept predictor for 150 epochs and the label predictor for 50 epochs. For training LaBo and V2C-CBM, we use the implementation of (Yang et al., 2023) with the same hyperparameters as discussed in the appendix of V2C-CBM. The baseline models are trained using PyTorch and transformers library (Wolf et al., 2020) with one NVIDIA RTX4090 Graphics card, and the inference of LLMs is conducted on NVIDIA L40 cards (1 card for LLaMA3-8B, Qwen2.5-7B, and Qwen2.5-14B, 2 cards for Qwen-2.5-32B, and 4 cards for Qwen2.5-72B and LLaMA3-70B).

## A.2    EVALUATION DETAILS OF CHAT-CBM

To control the output format of LLMs, we employ in-context examples and task instructions. The expected response format is <analysis: > <answer: class name>. Therefore, we determine whether the LLM provides a correct answer by directly matching <answer: target class name> within the LLM-generated response. We also checked the output of the model in advance and found that this kind of format requirement could be easily followed by LLMs. When prompting LLMs with the transformers library, we set the following hyperparameters for all LLMs: max_length=8192, do_sample=true, and top_k=10. We use left padding for the tokenizers and also set max_length=8192. But for intervention experiments, we set the max_length=10240 because the input length may exceed 8192 after several turns of intervention.

## A.3    TOP-N CLASSIFICATION ACCURACY OF CBMS

Since we use a standard independent CBM to generate the class label candidates, we provide the top-$k$ classification accuracy of the CBMs we used, and this serves as the upper bound of our Chat-CBM when no intervention is conducted. The results are presented in Table 6. We can see that though the concept prediction of CBM is not perfect and the large language model is not fine-tuned on the target tasks, our Chat-CBM can still approach the theoretical upper limit of performance, which demonstrates the potential of our method.

Table 6: Top-N classification accuracy of CBMs on datasets with concept label. (ResNet-101 as backbone)

| Dataset | CUB | | | AwA2 | | | WBC | | | |
|---|---|---|---|---|---|---|---|---|---|---|
| Top-N | 1 | 2 | 3 | 1 | 2 | 3 | 1 | 2 | 3 | 4 |
| CBM | 0.752 | 0.825 | 0.848 | 0.923 | 0.969 | 0.978 | 0.988 | 0.992 | 0.998 | 1.000 |
| Chat-CBM (LLaMA3-70B-Instruct) | 0.815 (N2-K2) | | | 0.964 (N2-K2) | | | 0.986 (N5-K3) | | | |

Table 7: Top-N classification accuracy of unsupervised CBMs on datasets without concept label.

| Dataset | DTD | | | | | Food-101 | | | | | Flower-102 | | | | |
|---|---|---|---|---|---|---|---|---|---|---|---|---|---|---|---|
| Top-N | 1 | 2 | 3 | 5 | 10 | 1 | 2 | 3 | 5 | 10 | 1 | 2 | 3 | 5 | 10 |
| LaBo | 0.769 | 0.877 | 0.916 | 0.950 | 0.997 | 0.924 | 0.968 | 0.981 | 0.986 | 0.991 | 0.993 | 0.998 | 0.999 | 0.999 | 1.000 |
| V2C-CBM | 0.782 | 0.896 | 0.933 | 0.960 | 0.999 | 0.927 | 0.970 | 0.982 | 0.989 | 0.994 | 0.987 | 0.996 | 0.998 | 0.999 | 1.000 |
| Dataset | CIFAR10 | | | | | CIFAR100 | | | | | ImageNet | | | | |
| Top-N | 1 | 2 | 3 | 5 | 10 | 1 | 2 | 3 | 5 | 10 | 1 | 2 | 3 | 5 | 10 |
| LaBo | 0.978 | 0.994 | 0.997 | 0.999 | 1.000 | 0.860 | 0.936 | 0.960 | 0.975 | 0.996 | 0.840 | 0.927 | 0.957 | 0.970 | 0.986 |
| V2C-CBM | 0.980 | 0.996 | 0.998 | 0.999 | 1.000 | 0.864 | 0.937 | 0.961 | 0.978 | 0.997 | 0.841 | 0.928 | 0.955 | 0.974 | 0.983 |

### A.4 Top-N Choice of Concepts for Unsupervised CBMs.

Both LaBo and V2C-CBM utilize a large number of concepts for prediction (typically $50 \times$ N_class). When these concepts are ranked by activation values, usually only the top 10 to 15 concepts are highly relevant to the image's semantics. Because Chat-CBM reasons on texts, the concepts with confusing semantics can have a noticeable negative impact. During development, we experimented with settings like top-5, 10, 15, 20 on small experiments. We found that the top-10's performance is the most robust. Therefore, this hyperparameter depends on the chosen concept extractor's performance, but might be determined by manually inspecting how many top concepts are generally relevant to the input image.

### A.5 Class Concept Semantics Prior Used for Different Datasets

Instead of using information from multiple places, such as Wikipedia, professional books, or websites, as prior knowledge. In the main experiments, we simply use the average concept of the class as the class concept semantics prior $\theta$. That is, we statistically calculate the probabilities of different concepts appearing in the current class based on the concepts and class labels in the training set, and construct the prior knowledge accordingly. The prior knowledge used for different datasets is detailed below.

- **CUB**: We directly utilize the average concept label for each class and select concepts with an occurrence probability greater than 0.5. For example, the prior knowledge for the black-footed albatross class includes: "bill shape is hooked seabird, underparts color is grey, breast pattern is solid, eye color is black, bill length is about the same as head, size is medium (9 - 16 in), back pattern is solid, tail pattern is solid, belly pattern is solid".
- **AwA2**: The concept labels for AwA2 are originally class-level; we directly use these as the class prior knowledge. For example, "antelope is usually associated with concepts including: furry, tough-skin, big, lean, hooves, longleg, tail, chewteeth, horns, walks, fast, strong, muscle, quadrapedal, active, agility, vegetation, forager, grazer, newworld, oldworld, plains, fields, mountains, ground, timid, group".
- **PBC**: We calculate the occurrence probability of each concept within each concept group for a given class in the training set. This is then used as the class prior knowledge. The specific representation is as follows: "for Lymphocyte: cell_size are mostly small, cell_shape is mostly round, nucleus_shape is mostly unsegmented-round, nuclear_cytoplasmic_ratio is high, chromatin_density is densely, cytoplasm_vacuole is no, cytoplasm_texture is clear, cytoplasm_color is light blue, granule_type is nil, granule_color is nil, granularity is no".
- **Datasets without concept labels**: Given the absence of ground-truth concept labels, we identify the 10 most frequently occurring concepts for each class based on the validation set images. These are then used as the class prior knowledge. Take Labo trained on the Flower-102 dataset as an example: "globe thistle is usually associated with concepts including: flower is also known as the blue thistle, thistle-like flower, attract bees, butterflies, and other pollinators, large, spiky, thistle-like flower, shaped like a thistle, self-seed itself, anti-inflammatory and healing properties, flower is also known as the bull thistle, not particularly attractive to bees or other pollinators, thistle is also known as the scotch thistle and is the national flower".

### A.6 Prompt and Output Format

Table 8: Prompt format for integrating prior knowledge on different classification tasks.

| Dataset | Format |
|---|---|
| CUB | "{classname} usually has: {concepts}" |
| AwA2 | "{classname} is usually associated with concepts including: {concepts}" |
| PBC | "for {classname}: {concepts[i]} is mostly / usually / (n%) ..." |
| Other datasets | "{classname} is usually associated with concepts including: {concepts}" |

## A.7 Details about Intervention on Unsupervised Datasets via Simulated Concept Correction or Removal

We employed a Multimodal Large Language Model (MLLM), specifically **Qwen2.5-VL-7B**, to simulate user interventions (concept correction or removal) on the unsupervised datasets.

The prompt provided to the MLLM for simulating these interventions included the following components:

1. The input image (`<image>`).
2. The analysis and prediction generated by the Chat-CBM.
3. The concept semantics utilized by the Chat-CBM.
4. The instruction for intervention selection, detailed as: "Based on the provided image, you are instructed to perform one of the following interventions:
   (a) Concept Correction (Emphasis): If a concept, present in the top-20 concepts, exists in the image but was ignored in the analysis text, emphasize it by outputting `<emphasize, concept>`.
   (b) Concept Removal: If a concept, present in the top-20 concepts, is incorrect based on the image, remove it by outputting `<remove, concept>`.

   In each iteration, you must select only one action and one concept, outputting in the format: `<action, concept>`."

After receiving the simulated intervention (type and concept) from the MLLM, we implement a preprocessing step. First, we validate that the concept generated by the MLLM does not contain any class names from the top-10 class candidates. If a class name is detected, the MLLM is prompted again until a valid concept is generated. Subsequently, the simulated intervention is converted into an explicit prompt for the downstream model:

- The "emphasize" action is converted into the prompt: "Concept exists in the image but was previously ignored. Consider the updated information and answer again."
- The "remove" action is converted into the prompt: "Concept was wrongly predicted and does not exist in the image. Ignore this concept and answer again."

In the next turn, the updated conversational history of Chat-CBM will replace the old conversational history and be provided to the MLLM again.

### A.8 Details about Intervention via Concept Augmentation

**Intervention under Controllable Incomplete Concept Settings.** We train independent CBMs on subsets of concepts from the CUB and AwA2 datasets. Specifically, we use 56/70/84/98/112 (full) concepts for CUB and 45/55/65/75/85 (full) concepts for AwA2, following the same hyperparameters described in Section A.1. These CBMs then serve as baseline models for the corresponding Chat-CBMs. Their classification performance of Chat-CBMs with LLaMA3-8B-Instruct is reported in Figure 6, where the starting point of each colored line is indicated by a triangle.

For experiments involving interventions with new concepts, we augment the concept space step by step. At each step, 14 new concepts are introduced for CUB (10 for AwA2), but only those overlapping with the ground-truth labels of the image are integrated. The resulting performance of Chat-CBMs is shown with circular points in Figure 6.

**Intervention using Wikipedia Descriptions.** For CUB and AwA2, we further collect class-level feature descriptions from Wikipedia and use them as additional concepts to intervene in Chat-CBMs. The intervention prompt is: *"In addition, we also know that <descriptions>. Answer again by considering the previous message and the new information."* To avoid class-label leakage, we replace the class name with general terms such as "the bird" (CUB) or "the animal" (AwA2). Two examples of such interventions are provided below.

- **black footed albatross**: The bird is a small member of the albatross family (while still large compared to most other seabirds) that has almost all black plumage. Some adults show white under

tail coverts, and all adults have white markings around the base of the beak and below the eye. As the birds age, they acquire more white at the base of the beak. Its beak and feet are also all dark. They have only one plumage. They measure 68 to 74 cm (27-29 in), have a wingspan of 190 to 220 cm (6.2-7.2 ft), and weigh 2.6 to 4.3 kg (5.7-9.5 lb). Males, at an average weight of 3.4 kg (7.5 lb), are larger than females, at an average of 3 kg (6.6 lb).

- **beaver**: The animals are the second-largest living rodents. The animals have large skulls with powerful chewing muscles. They have four chisel-shaped incisors that continue to grow throughout their lives. The incisors are covered in a thick enamel that is colored orange or reddish-brown by iron compounds. The lower incisors have roots that are almost as long as the entire lower jaw. Animals have one premolar and three molars on all four sides of the jaws, adding up to 20 teeth. The molars have meandering ridges for grinding woody material. The eyes, ears, and nostrils are arranged so that they can remain above water while the rest of the body is submerged. The nostrils and ears have valves that close underwater, while nictitating membranes cover the eyes.

### A.9    THE METADATA FOR LINE PLOT

The numerical metadata of the Figure 4 for intervention experiments on the supervised datasets is detailed as follows in Table 9, 10, and 11 respectively. The numerical metadata for the simulated user intervention experiments on unsupervised datasets in Figure 5 is presented in Table 12. And the metadata for intervention using new concepts in Figure 6 is provided in Table 13.

### A.10    USAGE OF LARGE LANGUAGE MODELS

During the preparation of this work, LLMs, including GPT-5 and Gemini, were used for checking syntax errors and polishing the writing.

Table 9: Metadata (classification accuracy) for the CUB dataset in Figure 4.

| model \ ratio | 0.0 | 0.1 | 0.2 | 0.3 | 0.4 | 0.5 | 0.6 | 0.7 | 0.8 | 0.9 | 1.0 |
|---|---|---|---|---|---|---|---|---|---|---|---|
| Hard CBM | 0.7076 | 0.7378 | 0.7672 | 0.7995 | 0.8533 | 0.8931 | 0.9308 | 0.9586 | 0.9838 | 0.9960 | 0.9983 |
| CBM | 0.7525 | 0.7846 | 0.8112 | 0.8460 | 0.8749 | 0.8918 | 0.9134 | 0.9326 | 0.9513 | 0.9694 | 0.9738 |
| ECBM | 0.8063 | 0.8133 | 0.8143 | 0.8194 | 0.8475 | 0.8702 | 0.9046 | 0.9263 | 0.9327 | 0.9549 | 0.9801 |
| CEM | 0.7991 | 0.8261 | 0.8472 | 0.9121 | 0.9303 | 0.9420 | 0.9593 | 0.9661 | 0.9803 | 0.9855 | 0.9911 |
| Chat-CBM | 0.7978 | 0.8886 | 0.9384 | 0.9617 | 0.9778 | 0.9874 | 0.9921 | 0.9943 | 0.9967 | 0.9975 | 0.9984 |
| SCBM | 0.7780 | 0.8574 | 0.9002 | 0.9301 | 0.9574 | 0.9753 | 0.9872 | 0.9931 | 0.9960 | 0.9974 | 1.0000 |
| CBM-MLP(1) | 0.7908 | 0.8319 | 0.8592 | 0.8837 | 0.9250 | 0.9457 | 0.9604 | 0.9782 | 0.9915 | 0.9975 | 0.9995 |
| CBM-MLP(2) | 0.7684 | 0.8019 | 0.8260 | 0.8664 | 0.8966 | 0.9254 | 0.9447 | 0.9673 | 0.9809 | 0.9961 | 0.9981 |

Table 10: Metadata (classification accuracy) for the AwA2 dataset in Figure 4.

| model \ ratio | 0.00 | 0.09 | 0.18 | 0.27 | 0.36 | 0.45 | 0.55 | 0.64 | 0.73 | 0.82 | 0.91 | 1.00 |
|---|---|---|---|---|---|---|---|---|---|---|---|---|
| Hard CBM | 0.9006 | 0.9351 | 0.9526 | 0.9575 | 0.9661 | 0.9721 | 0.9766 | 0.9854 | 0.9914 | 0.9932 | 0.9950 | 0.9950 |
| CBM | 0.9233 | 0.9300 | 0.9475 | 0.9635 | 0.9728 | 0.9786 | 0.9826 | 0.9894 | 0.9939 | 0.9950 | 0.9974 | 0.9974 |
| ECBM | 0.9160 | 0.9217 | 0.9266 | 0.9375 | 0.9491 | 0.9773 | 0.9864 | 0.9932 | 0.9951 | 0.9961 | 0.9973 | 0.9973 |
| CEM | 0.9242 | 0.9411 | 0.9583 | 0.9727 | 0.9831 | 0.9947 | 0.9982 | 0.9989 | 0.9992 | 0.9995 | 0.9996 | 0.9996 |
| Chat-CBM | 0.9572 | 0.9732 | 0.9805 | 0.9856 | 0.9881 | 0.9930 | 0.9947 | 0.9954 | 0.9962 | 0.9989 | 0.9996 | 0.9996 |
| SCBM | 0.9170 | 0.9531 | 0.9712 | 0.9805 | 0.9881 | 0.9930 | 0.9973 | 0.9992 | 1.0000 | 1.0000 | 1.0000 | 1.0000 |
| CBM-MLP(1) | 0.9448 | 0.9488 | 0.9633 | 0.9694 | 0.9773 | 0.9833 | 0.9880 | 0.9927 | 0.9981 | 0.9989 | 1.0000 | 1.0000 |
| CBM-MLP(2) | 0.9433 | 0.9472 | 0.9631 | 0.9674 | 0.9756 | 0.9823 | 0.9877 | 0.9920 | 0.9974 | 0.9992 | 1.0000 | 1.0000 |

Table 11: Metadata (classification accuracy) for the PBC dataset in Figure 4.

| model \ ratio | 0.00 | 0.07 | 0.14 | 0.21 | 0.29 | 0.36 | 0.43 | 0.50 | 0.57 | 0.64 | 0.71 | 0.79 | 0.86 | 0.93 | 1.00 |
|---|---|---|---|---|---|---|---|---|---|---|---|---|---|---|---|
| Hard CBM | 0.9593 | 0.9606 | 0.9678 | 0.9681 | 0.9684 | 0.9808 | 0.9812 | 0.9804 | 0.9806 | 0.9810 | 0.9823 | 0.9854 | 0.9871 | 0.9875 | 0.9883 |
| CBM | 0.9880 | 0.9837 | 0.9834 | 0.9804 | 0.9804 | 0.9831 | 0.9819 | 0.9782 | 0.9780 | 0.9781 | 0.9791 | 0.9802 | 0.9805 | 0.9802 | 0.9799 |
| ECBM | 0.9941 | 0.9931 | 0.9944 | 0.9945 | 0.9945 | 0.9955 | 0.9962 | 0.9962 | 0.9966 | 0.9973 | 0.9981 | 0.9989 | 0.9991 | 0.9993 | 0.9994 |
| CEM | 0.9933 | 0.9943 | 0.9955 | 0.9967 | 0.9967 | 0.9977 | 0.9975 | 0.9974 | 0.9981 | 0.9983 | 0.9985 | 0.9987 | 0.9995 | 0.9996 | 0.9998 |
| Chat-CBM | 0.9764 | 0.9803 | 0.9822 | 0.9911 | 0.9928 | 0.9941 | 0.9956 | 0.9966 | 0.9971 | 0.9972 | 0.9979 | 0.9983 | 0.9988 | 0.9981 | 0.9989 |
| SCBM | 0.9924 | 0.9920 | 0.9920 | 0.9920 | 0.9941 | 0.9933 | 0.9933 | 0.9933 | 0.9920 | 0.9933 | 0.9941 | 0.9920 | 0.9903 | 0.9890 | 0.9890 |
| CBM-MLP(1) | 0.9954 | 0.9939 | 0.9939 | 0.9939 | 0.9939 | 0.9939 | 0.9926 | 0.9897 | 0.9890 | 0.9890 | 0.9897 | 0.9897 | 0.9906 | 0.9900 | 0.9890 |
| CBM-MLP(2) | 0.9945 | 0.9935 | 0.9935 | 0.9935 | 0.9936 | 0.9935 | 0.9913 | 0.9903 | 0.9910 | 0.9913 | 0.9913 | 0.9906 | 0.9906 | 0.9903 | 0.9890 |

Table 12: Metadata (classification accuracy) for Figure 5. N. denotes the number of interventions.

| Dataset \ N. | Chat-CBM (V2C-CBM) | | | | | | Chat-CBM (LaBo) | | | | | |
|---|---|---|---|---|---|---|---|---|---|---|---|---|
|  | 0 | 1 | 2 | 3 | 4 | 5 | 0 | 1 | 2 | 3 | 4 | 5 |
| CIFAR10 | 0.955 | 0.967 | 0.982 | 0.991 | 0.996 | 0.999 | 0.889 | 0.902 | 0.945 | 0.959 | 0.976 | 0.983 |
| CIFAR100 | 0.727 | 0.803 | 0.853 | 0.891 | 0.925 | 0.943 | 0.670 | 0.751 | 0.843 | 0.898 | 0.921 | 0.930 |
| Food-101 | 0.786 | 0.873 | 0.924 | 0.947 | 0.958 | 0.976 | 0.753 | 0.855 | 0.907 | 0.932 | 0.946 | 0.965 |
| Flower-102 | 0.914 | 0.966 | 0.986 | 0.992 | 0.996 | 0.999 | 0.876 | 0.960 | 0.980 | 0.991 | 0.998 | 0.999 |
| DTD | 0.734 | 0.842 | 0.906 | 0.928 | 0.941 | 0.961 | 0.677 | 0.811 | 0.893 | 0.932 | 0.950 | 0.974 |

Table 13: Metadata for Figure 6. N. denotes the number of concepts. We use start=1,2,3,4,5 to represent the corresponding starting number of concepts for the CUB and AwA2 datasets.

| start= \ N. | CUB | | | | | | AwA2 | | | | | |
|---|---|---|---|---|---|---|---|---|---|---|---|---|
|  | 56 | 70 | 84 | 98 | 112 | +wiki | 45 | 55 | 65 | 75 | 85 | +wiki |
| 1 | 0.620 | 0.685 | 0.729 | 0.769 | 0.805 | - | 0.901 | 0.924 | 0.943 | 0.950 | 0.959 | - |
| 2 | - | 0.675 | 0.736 | 0.798 | 0.858 | - | - | 0.920 | 0.945 | 0.953 | 0.960 | - |
| 3 | - | - | 0.727 | 0.817 | 0.893 | - | - | - | 0.934 | 0.954 | 0.961 | - |
| 4 | - | - | - | 0.760 | 0.912 | - | - | - | - | 0.949 | 0.967 | - |
| 5 | - | - | - | - | 0.797 | 0.945 | - | - | - | - | 0.957 | 0.989 |

# B  ADDITIONAL EXPERIMENTS

## B.1  INTERVENTION VIA HIGH-LEVEL STRATEGY GUIDANCE

We conduct supplementary experiments on the SkinCon dataset (Daneshjou et al., 2022), a skin disease dataset annotated with expert-defined concepts, to investigate Chat-CBM's generalization ability to domains where the frozen pretraining corpus offers less specialized prior knowledge. Furthermore, the established ABCDE (stands for "Asymmetrical, Border, Color, Diameter, Evolving") skin assessment strategy (Robinson & Turrisi, 2006; De Giorgi et al., 2013; Duarte et al., 2021) can serve as a high-level reasoning strategy, providing an effective and clinically relevant method to evaluate Chat-CBM's intervention capabilities for receiving strategy guidance.

We use the SkinCon dataset with 22 concepts following (Pang et al., 2024), including: Papule, Plaque, Pustule, Bulla, Patch, Nodule, Ulcer, Crust, Erosion, Atrophy, Exudate, Telangiectasia, Scale, Scar, Friable, Dome-shaped, Brown(Hyperpigmentation), White(Hypopigmentation), Purple, Yellow, Black, Erythema. We train an independent CBM with an ImageNet-pretrained ResNet-50 as the backbone, and an MLP with one hidden layer (dimension 64) as the label predictor to classify images into two classes (benign, malignant). All the other training details are the same in appendix A.1. We implement Chat-CBM upon the baseline CBM with LLaMA3-8B-Instruct. The performance of the CBM and Chat-CBM is presented in Table 14.

Table 14: Classification accuracy on the SkinCon dataset.

| Metric | CBM | Chat-CBM | Chat-CBM + ABCDE |
|---|---|---|---|
| Concept Accuracy | 0.883 (same concept predictor) | | |
| Class Accuracy | 0.671 | 0.701 | 0.790 |

For evaluating the capability of using high-level strategy guidance as a test-time intervention, we designed a strategy grounded in the ABCDE skin assessment rule. Specifically, we instructed the model with the following prompt:

*Please utilize the ABCDE (Asymmetry, Border, Color, Diameter, Evolving) rule to re-evaluate malignancy risk. Map specific features such as Color tags (e.g., Black, Brown, White, Erythema) to the 'C' criterion for variegation, and structural changes (e.g., Ulcer, Friable, Crust) to the 'E' criterion for aggression to distinguish between high-risk malignant profiles and benign inflammatory patterns. Explicitly evaluate the lesion's risk level based on these indicators.*

Following the application of this high-level guidance, classification accuracy increased from 0.701 to 0.790, which validates the effectiveness of the test-time interventions via high-level strategy guidance supported by Chat-CBM.

## B.2  USER STUDY ON MODEL INTERVENTION

Chat-CBMs are primarily developed to explore the intervention capabilities enabled by language-based label predictors. Following previous work (Chauhan et al., 2023) and (Zarlenga et al., 2023), we first quantified intervention effectiveness using the intervention-accuracy curves presented in Figures 4, 5, and 6 in the main body of the paper. These automated results demonstrate the intervention efficacy of our Chat-CBMs.

To further validate the practical utility and intuitive interactivity of Chat-CBMs for end-users, we developed a Gradio-based interactive interface and conducted a supplementary small-scale user study on the CUB dataset.

**Participants**   We recruited $N = 10$ participants from a single research institution. Participants were randomly selected and lacked familiarity with the CBM domain and professional ornithological knowledge, ensuring the findings reflect general user experience and intuitive usability. They were divided into two groups: the **Baseline CBM Group** ($N = 5$) and the **Chat-CBM Group** ($N = 5$).

**Task and Materials**   We randomly selected 100 test samples from the CUB test set, balanced according to the base model's prediction results (correct/incorrect). These were divided into five unique intervention sets of 20 images each. Each participant received one set comprising:

1. 10 cases where the model's initial prediction was incorrect.
2. 10 cases where the model's initial prediction was correct.

Users were tasked with correcting or confirming model predictions for blinded test images; users did not see the ground truth label for either the concepts or the class labels. Intervention could be stopped when the user was satisfied with the resulting prediction or chose to cease the effort. The user interfaces for the two intervention modes are illustrated in Figure 7 (Baseline CBM) and Figure 8 (Chat-CBM).

1. **UI for the Baseline CBM**: As shown in Figure 7, the interface allows users to upload an image and trigger model inference. The visualization includes the top-3 class predictions (top right), concept contributions in descending order (top right), and the predicted probability of each concept (bottom left). The concept probability component also offers sorting functionalities (alphabetical, probability descending/ascending) to facilitate rapid identification of concepts for intervention.
2. **UI for the Chat-CBM**: As shown in Figure 8, this interface includes the same concept prediction and numerical editing components as the baseline. The key feature is the language-based inference process of the Chat-CBM, visualized at the top right, where users submit their interventions via text in the dialogue box.

We utilized `LLaMA3-8B-Instruct` as the language-based label predictor for Chat-CBM in this user study.

**Evaluation Metrics**   We recorded three metrics to evaluate the efficiency, efficacy, and user experience of the intervention mechanisms, collecting 100 intervention data points for each group:

1. **Intervention Time**: The time taken to complete the intervention task per image. The start time was recorded upon image upload, and the end time was recorded when the user proceeded to the next task (uploading a new image, clearing the image, or clearing the conversational history).
2. **Success Rate**: The correctness of the final model prediction after human guidance.
3. **User Satisfaction**: A binary (1/0) rating indicating whether the user was "satisfied with the prediction explanation and intervention control."

**Results**   We collected 100 user intervention data points from each group, with results presented in Table 15. These findings suggest that Chat-CBM enhances interactivity and efficiency for end-users.

The most substantial speedup arose because natural language enabled users to more quickly pinpoint the concept requiring modification, directly leveraging the model's analysis and the visual cues in the image. In the standard CBM scenario, users incur significant cognitive overhead by manually inspecting concept probabilities and contributions to determine if a specific, modifiable concept exists. Figure 8 illustrates a typical scenario where a user inputs "the bill of the bird is yellow instead of black." Crucially, the concept `bill color::yellow` may not exist within the CUB's 112 concepts. Chat-CBM's language interface provides a more intuitive and forgiving path, circumventing this conceptual mapping difficulty. This direct and effective intervention capability also likely contributed to higher user satisfaction.

While Chat-CBM achieved a higher overall success rate than the baseline CBM (consistent with the better automated intervention efficacy demonstrated in the main body), we observed a difference in robustness to interventions. The standard CBM proved more resistant to "unprofessional" human interventions; for instance, incorrectly deactivating a correct concept often had a negligible effect on the final prediction. Conversely, the high sensitivity of the Chat-CBM's language predictor led to 4 cases where an initially correct prediction became incorrect post-intervention, a situation that did not occur in the baseline CBM group. This highlights a sensitivity trade-off.

We acknowledge the limitations of this small-scale user study, particularly the potential for participant bias given the non-random sampling of $N = 10$ participants from a single institute. Nevertheless, these results, serving as a qualitative supplement to our extensive automated intervention experiments, collectively demonstrate the strong potential of language-based label predictors to enable more intuitive and effective user interventions for CBMs.

Table 15: Results of the user intervention study.

| Metric | Intervention Time | Success Rate | User Satisfaction |
|---|---|---|---|
| CBM | 215.36 seconds | 0.63 | 0.77 |
| Chat-CBM | 100.62 seconds | 0.82 | 0.84 |

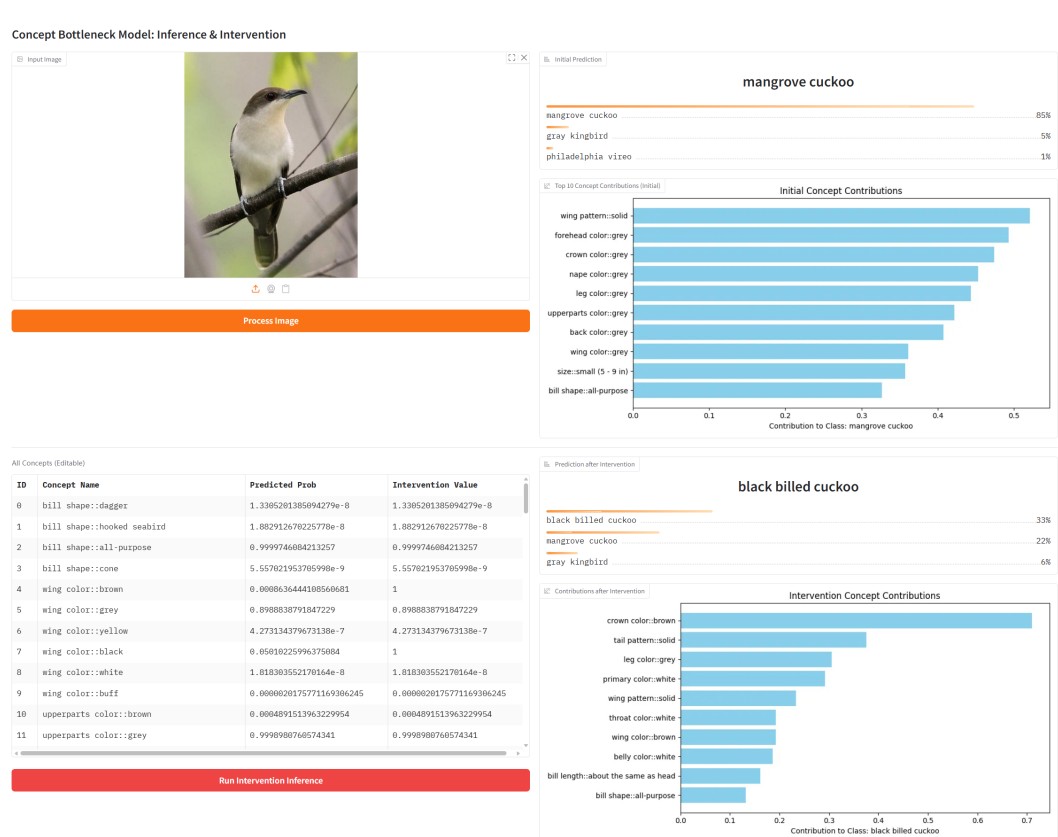

Figure 7: The user interface of user intervention for the standard CBM models. Users can upload an image, view the initial prediction result, which includes class prediction, concept prediction, and the concept contribution to the class prediction. After the initial prediction, the user can also edit the concept predictions to re-infer the class label of the image. The class label of the bird in this example image is *black billed cuckoo*.

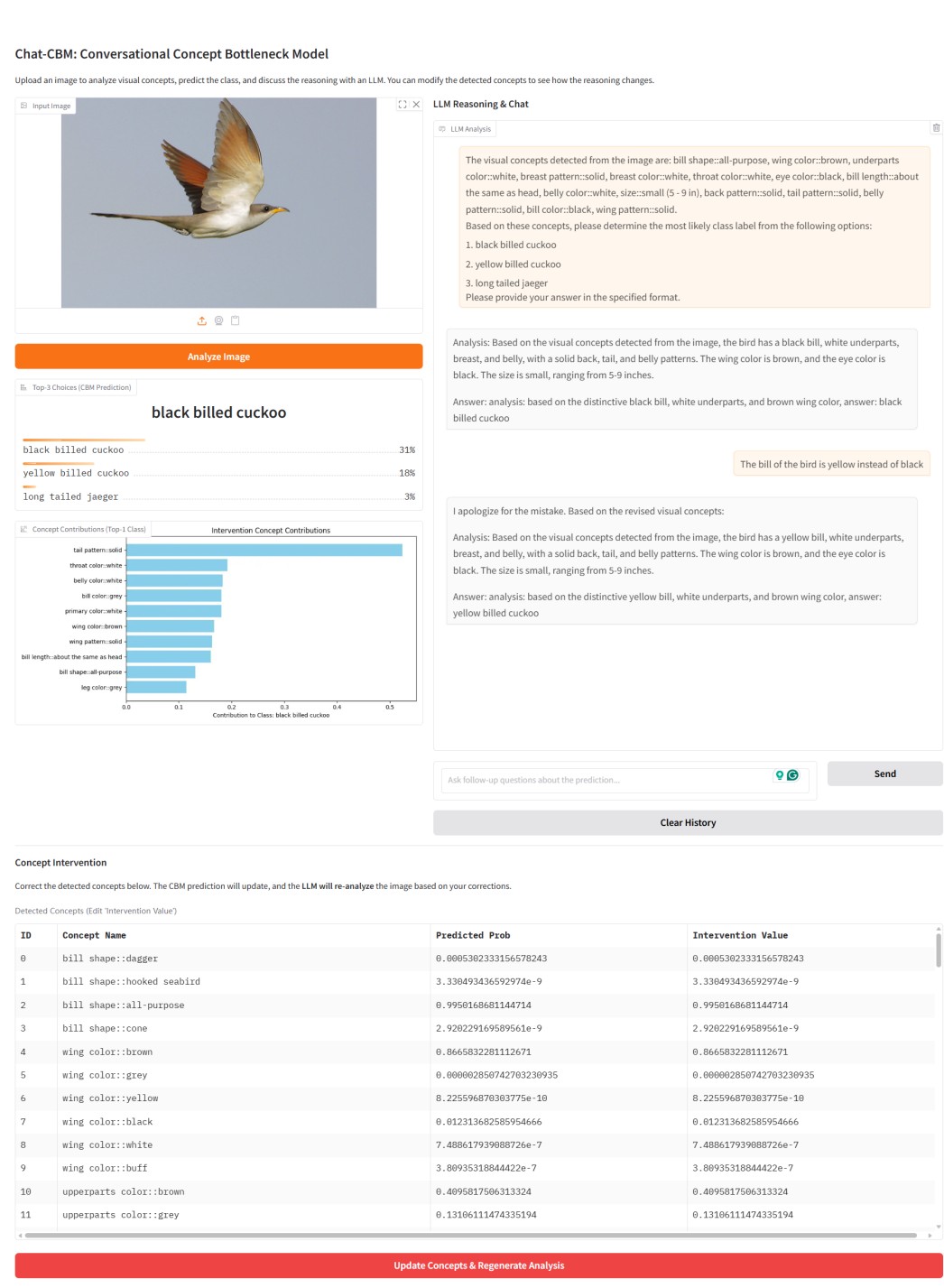

Figure 8: The user interface of user intervention for Chat-CBMs. Users can upload an image, view the initial prediction result, which includes class prediction, concept prediction, and the concept contribution to the class prediction of the baseline CBMs. And also the analysis and prediction generated by the LLM. After the initial prediction, the user can also edit the concept predictions to re-infer the class label of the image, or send text to add new concepts/remove concepts/new information and so on. The class label of the bird in this example image is *yellow billed cuckoo*.

## C  MORE VISUALIZATION RESULTS AND FAILURE CASE ANALYSIS

### C.1  SUCCESSFUL CASES.

We provide more examples of the reasoning process or the intervention process of Chat-CBMs in Figure 9, 10, and 11 for datasets with concept labels. And Figure 12 and Figure 13 are for datasets without concept labels. In these successful cases, Chat-CBMs naturally exhibit both positive (focus on the presence of distinctive concepts) and negative reasoning processes (focus on the absence of distinctive concepts) on top of the concepts.

### C.2  FAILURE CASES AND ANALYSIS.

However, beyond the intuitive failure cases, when the concept prediction results are noisy, which mislead the understanding of the LLM. There are cases when Chat-CBMs fail to give the right predictions based on mostly accurate concepts or fail to correct their predictions with user interventions. Here, we visualize and analyze several most common cases of Chat-CBM failures.

Figure 11 shows that although the concept predictors output mostly accurate concepts (only one concept "longneck" is wrongly predicted), which may have little influence on the baseline CBMs, a single wrong concept with a distinctive semantic will lead to wrong class label prediction for Chat-CBMs. For example, since the concept semantics of "longneck" is distinctive for differentiating deer and antelope based on the concept sets, Chat-CBM predicts it as deer based on "*In the previous cases where longneck was present, the classification was deer. Therefore, even though many other attributes like furry, toughskin, big, lean overlap between the two, the presence of longneck suggests a deer*". But we think that this failure case also indicates that language-based label predictors are more faithful to the concept semantics, which means they're more aligned with human understanding of the concept-based inference process.

Also, since we use the baseline CBM to generate the class candidates, and if there is no answer in the class candidates, Chat-CBM can also not change this (as shown in Figure 14). So the top-k accuracy of the baseline model determines the upper bound of the Chat-CBM's performance.

Another common situation for both the failures of inference and interventions is that the incomplete concept sets and abstract (or vague) concept semantics may mislead the understanding of the LLM of the image. And this is also related to the limitation of Chat-CBM for having no awareness of concept uncertainty and absence, as we discuss more in the appendix D. Figure 15 presents such a situation where the concept semantics of cow and ox are mostly the same for many cases in the AwA2 dataset, and Chat-CBM doesn't have the awareness of concept uncertainty, which may be used by the baseline models to classify cow and ox.

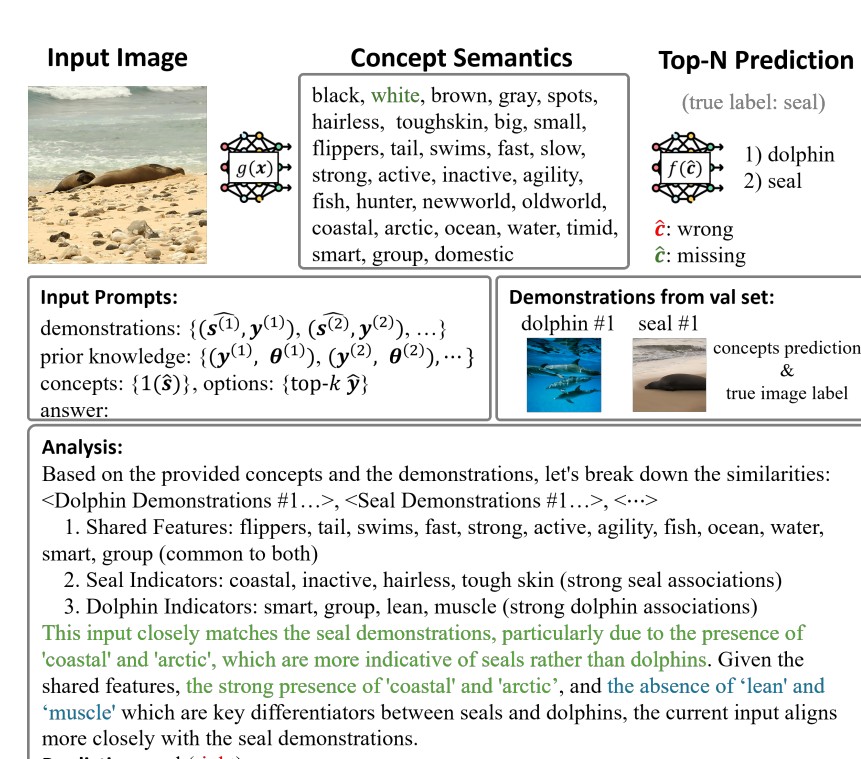

Figure 9: Visualization of the inference process of Chat-CBM on the AwA2 dataset. Green highlights the positive reasoning and blue highlights the negative reasoning process.

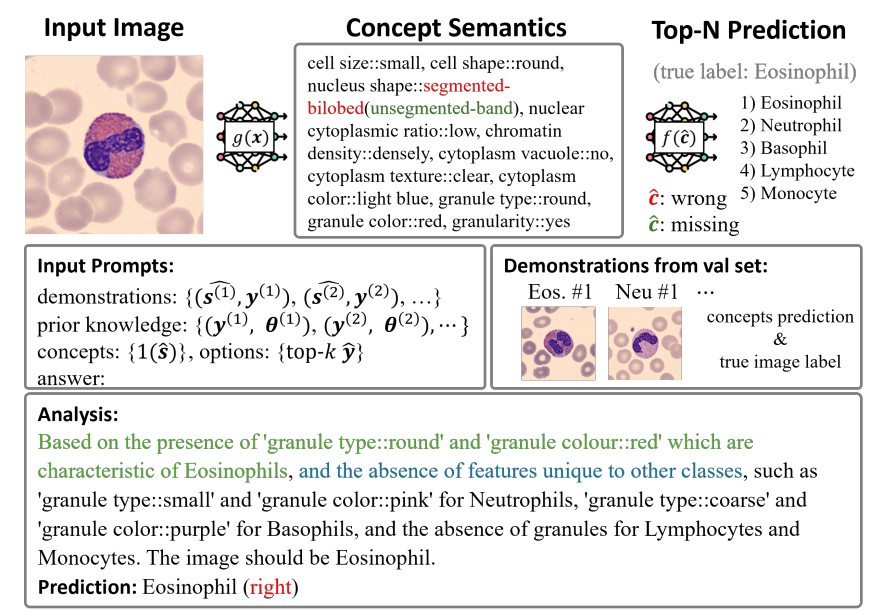

Figure 10: Visualization of the inference process of Chat-CBM on the PBC dataset. Green highlights the positive reasoning and blue highlights the negative reasoning process.

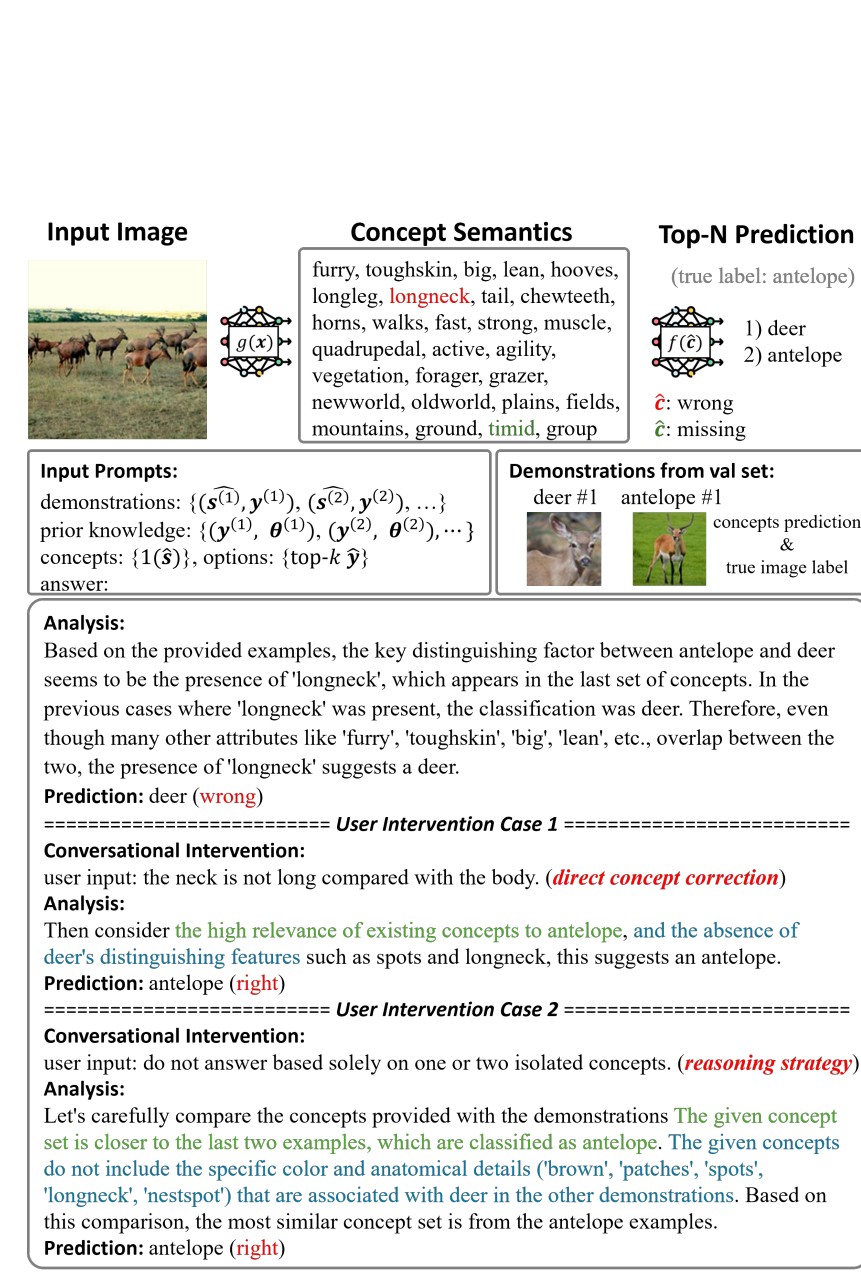

Figure 11: Visualization of the conversational intervention process of Chat-CBM on the AwA2 dataset. Green highlights the positive reasoning and blue highlights the negative reasoning process. Users can either directly correct concept predictions like standard CBMs (**case 1**) or give a high-level reasoning strategy to guide thinking (**case 2**).

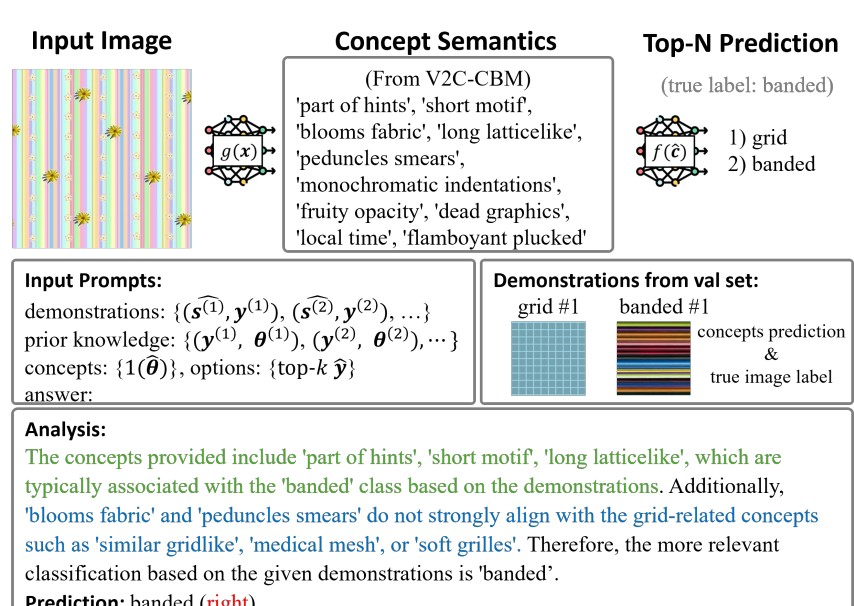

Figure 12: Visualization of the inference process of Chat-CBM on the DTD dataset. The concept bank is from V2C-CBM.

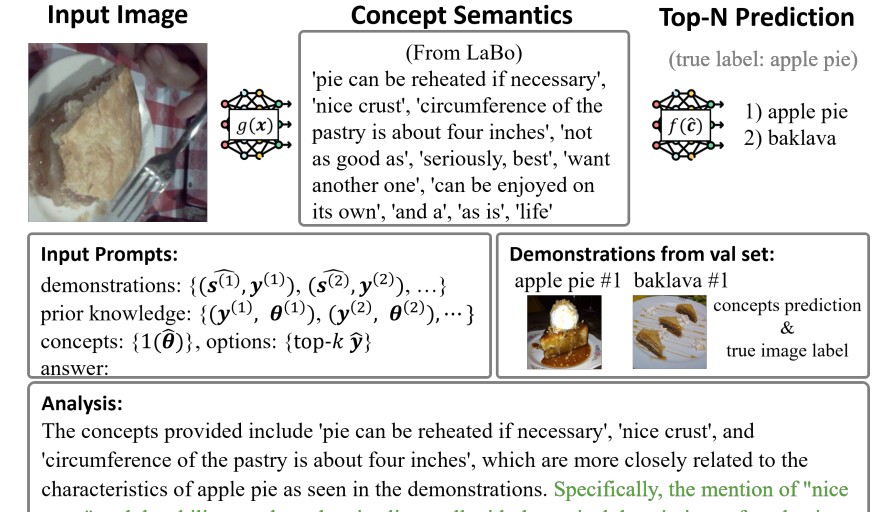

Figure 13: Visualization of the inference process of Chat-CBM on the Food-101 dataset. The concept bank is from LaBo.

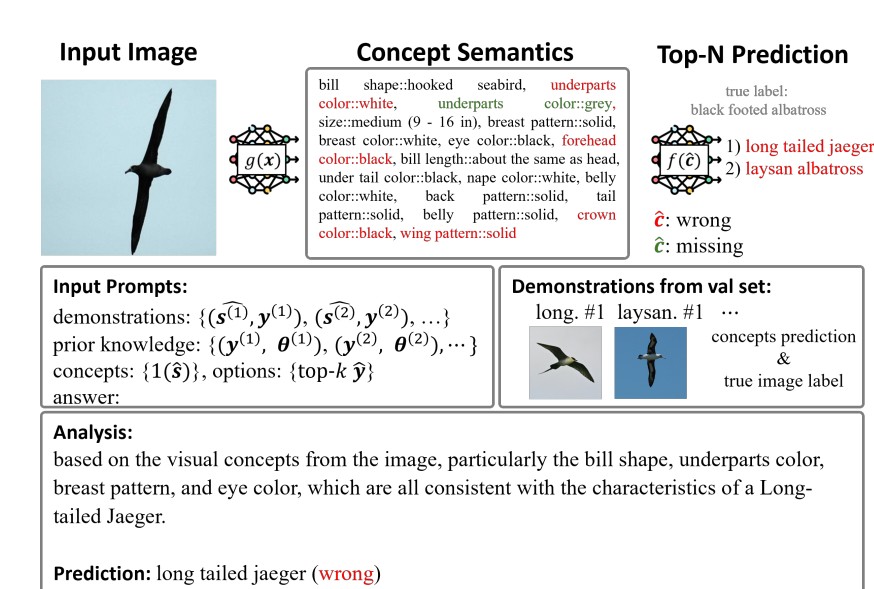

Figure 14: Failure case: the top-k prediction of the baseline model doesn't contain the right class candidate, and Chat-CBM can not handle such a situation, which leads to wrong prediction.

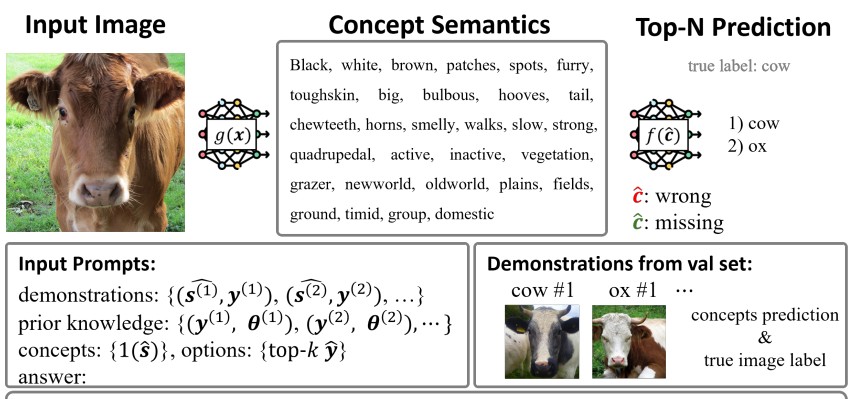

Figure 15: Failure case: the concept semantics of cow and ox are mostly the same for many cases in the AwA2 dataset, and Chat-CBM doesn't have the awareness of concept uncertainty, which may be used by the baseline models to classify cow and ox. This is also related to the design choice of Chat-CBM, which prevents the possible concept leakage problem where the concept neuron actually encodes some other information, but may also decrease the performance of Chat-CBM in such cases.

# D LIMITATIONS AND FUTURE WORK

**Hard to learning additional information under incomplete concept supervision.** Chat-CBMs leverage the concept semantics for reasoning, so they inherently prevent possible concept leakage problems of existing CBMs, in which some concept activations may actually serve as a class label proxy. But the structure of Chat-CBM also makes it challenging to learn additional information to improve performance when concept label discriminability is very insufficient, as explored in Concept Embedding Models (CEM) (Zarlenga et al., 2022).

However, in cases where only a portion of the concept set is incomplete (which we think is a more common case), Chat-CBM's strong generalization and few-shot capabilities will still allow its performance to be comparable to or better than baselines. To validate this, we train independent CBMs using reduced subsets of the original concept labels (56/70/84/98/112 for CUB; 45/55/65/75/85 for AwA2), and then evaluate them. The results in Table 16 and Table 17 validate that Chat-CBMs still achieve better performance compared to CBMs under this setting.

Table 16: Performance of CBMs and Chat-CBMs under incomplete concepts on the CUB dataset.

| Number of Concepts | 56 | 70 | 84 | 98 | 112 |
|---|---|---|---|---|---|
| Independent CBMs | 0.591 | 0.666 | 0.712 | 0.743 | 0.752 |
| Chat-CBMs (LLaMA3-8B-Instruct) | 0.620 | 0.675 | 0.727 | 0.760 | 0.797 |

Table 17: Performance of CBMs and Chat-CBMs under incomplete concepts on the AwA2 dataset.

| Number of Concepts | 45 | 55 | 65 | 75 | 85 |
|---|---|---|---|---|---|
| Independent CBMs | 0.913 | 0.915 | 0.921 | 0.922 | 0.923 |
| Chat-CBMs (LLaMA3-8B-Instruct) | 0.901 | 0.920 | 0.934 | 0.949 | 0.957 |

**Unawareness of Concept Uncertainty and Absence.** Because we directly use a threshold to decode concept probabilities into concept semantics, the language-based label predictor currently has no awareness of the concept uncertainty, which may omit useful information that the concept predictor's confidence can provide. Also, although Chat-CBMs naturally exhibit both positive and negative reasoning processes as shown in Figure 3, 9, 10, 11, 12, and 13, the language-based label predictor will not have explicit knowledge about the concept's absence if we don't provide the whole set of possible concepts in the prompt. In this work, we mainly explore the intervention capabilities that can be provided by the language-based predictors (LLMs), but designing better concept extraction and representation methods for language-based label predictors is worth exploring in future work.

**Computational Cost.** Using an LLM as a language-based classifier typically means an extra 10+ 100+ GBs of GPU memory per image (depending on the LLM size), and an average of 3.32 sec/image for generating complete outputs (until the EOS token) on an NVIDIA L40 GPU using LLaMA-3-8B-Instruct. While this does increase computational cost and latency for large-scale experiments (e.g., testing Chat-CBM performance on new benchmarks), it's acceptable for user-facing interactive use cases. The streaming output style, the widespread availability, and the rapid response of LLM APIs can further mitigate this influence. For large-scale deployment, given that we retain the explicit concept bottleneck structure in Chat-CBMs, we can also cache common concept input contexts during the model's actual service phase, thereby reducing operational costs and improving inference speed. However, there is no denying that Chat-CBM does introduce significant computational overhead compared to the standard CBM architecture.

**Potential Harmful Knowledge in LLMs.** As Chat-CBM builds on frozen LLMs as label predictors, it inevitably inherits the knowledge embedded in these models. While this enables strong semantic reasoning, it also carries potential risks: LLMs may encode harmful, biased, or misleading knowledge, which could in turn affect the model's predictions or the interaction process. Although our experiments are limited to benchmark datasets and do not involve deployment in high-stakes applications, these

risks must be considered before applying Chat-CBM in sensitive domains such as medicine or law. Future work should incorporate alignment strategies (e.g., safety-tuned LLMs (Wu et al., 2025b), safeguarding (Wei et al., 2023; 2025), or unlearning the harmful knowledge (Wu et al., 2025a)) to ensure that user-facing interventions remain safe, unbiased, and reliable.

