# OpenReview forum: "Chat-CBM: Towards Interactive Concept Bottleneck Models with Frozen Large Language Models"
_ICLR.cc/2026/Conference — Submitted to ICLR 2026_

### Official Review · Reviewer_8iQG · 2025-10-17

**Soundness:** 3
**Presentation:** 4
**Contribution:** 1
**Rating:** 2
**Confidence:** 4

**Summary:**

Chat-CBM proposes an integration of LLMs into the CBM framework by adopting a semantic bottleneck (i.e. textual description of the concepts) which is consequently passed through the LLM. As such, the LLM replaces the target classifier. This framework enables a lot of flexibility for interventions, where they are not bound to the concept set of the concept encoder, and can interact with a user directly with language, which is a benefit. The LLM is queried with two important additions: (i) Class Concept Semantics Prior Integration, which captures the class-level concept structure that is present in e.g. CUB, and thereby gives the LLM context about the data structure, and (ii) In-context learning, which gives the LLM data examples (incl. predicted concepts) for each candidate class, thereby aiding the model in understanding potential shifts of the concept encoder.
Experiments are very extensive for datasets with ground-truth concepts and discovered concepts.

**Strengths:**

Since the era of LLMs, multiple works have tried to leverage them in combination with CBMs. The direct usage of LLMs as target predictor is neat, as it allows for a lot of flexibility at the intervention stage. As such, I see the main contributions of this work to be the smart setup of the LLM prompting for giving meaningful context about the task, as well as the design choice of using LLMs directly, thereby enabling flexible intervenability. One additional strength is that for user interventions, it is more convenient for users to intervene in language space rather than numerical concept space, thereby reducing the adoption barrier of Chat-CBM.

The paper is nicely written, clearly explained, formalized, and outlined. The experiments are impressively vast, spanning supervised and unsupervised datasets, interventions, and ablations.

**Weaknesses:**

To summarize my main critique in one sentence: I see this work as mainly adapting LLM terminology to CBM terminology.

The usage of LLMs and in-context learning for classification is common[1]. Similarly, the text interventions are the same thing as having a normal conversation with a chatbot; a direct consequence of using LLMs.
As I see it, the main novelty of this work is adapting few-shot LLMs to fit a CBM framework by smart prompt design that incorporates dataset priors. While an art in itself, I do not think great prompt engineering warrants acceptance.

Beyond the aforementioned main point, I do have additional critiques:
* The common reason for adopting a linear classification head is its interpretability, since the mapping from concepts to target are clear. By replacing this layer with a LLM, the reasoning becomes opaque. While a linear layer indicates directly that the concepts are being used for prediction, which is the desired behavior, the few shot LLM seems to rather perform a prototype-based prediction from the in-context examples, but it is unclear if that is being done and what contributes how (note that I do not think an LLM's reasoning output is interpretability[2]).
* I do not fully agree with some of the conclusions drawn from the results. (i) For accuracy on concept-supervised datasets, ChatCBM is compared to baselines that employ a linear head. As such, I wonder if the performance gain from ChatCBM is not merely due to the introduction of a non-linear classifier that can better model the concept--class mapping. An effect that could also be achieved by using an MLP instead of a linear layer. Also, evaluating a 70B LLM versus a linear layer does not seem like a fair comparison for performance.
(ii) For accuracy on concept-unsupervised datasets, the authors argue that a fair comparison would be to compare 2-shot ChatCBM with 2shot baselines. I disagree. To me, it the 2-shot limitation of ChatCBM is an inherent downside of the method's design. As such, the other methods do not need to be artificially worsened, if they do not have this limitation. Comparing 2shot ChatCBM to allshot baselines leads to worse performance of ChatCBM. (iii) Intervention on unsupervised datasets: The assistant LLM seems the ground truth class label, which is an unfair advantage. If a user had access to the class label, there would not be any reason for intervening on the model. An alternative setup might be to provide the assistant LLM the ground truth knowledge for the concepts. Alternatively, standard intervention policies such as random / uncertainty-based could be employed, where no LLM is required. (iv) Similarly, for the wikipedia example, using the class's description, rather than what is visible from the image, is not a realistic intervention setting.
* It is stated that the unsupervised methods struggle from a noisy dataset, which I agree with. However, given that ChatCBM also bases on these methods' concept annotations for their approach, they naturally inherit all these issues instead of solving them.
* No code is provided, thereby limiting reproducibility.
* The semantic bottleneck of ChatCBM does not take into account the predicted concept uncertainty, thereby omitting valuable information for its considerations. Also, absent concepts (i.e. c=0) are omitted instead of being added to the prompt as being absent.
* The flexibility to add new concepts for interventions at test time is not new. For example, [3] targets exactly this problem.
* It is unclear whether there might be some test set leakage due to the LLM likely being trained on it.


[1] Brown, Tom, et al. "Language models are few-shot learners." Advances in neural information processing systems 33 (2020): 1877-1901.

[2] Barez, Fazl, et al. "Chain-of-thought is not explainability." Preprint, alphaXiv (2025): v1.

[3] Laguna, Sonia, et al. "Beyond concept bottleneck models: How to make black boxes intervenable?." Advances in neural information processing systems 37 (2024): 85006-85044.

**Questions:**

I invite the authors to respond to any of the previously stated weaknesses and potential misunderstandings. In addition, I have a few minor questions

* Will reproducible code be provided?
* In App. A.1, the SCBM model is stated as a baseline, but it does not appear in the results. Can the authors add it to the results or explain why they are not showing it?
* Contrary to the statement on p.4 "The top-N choice (of unsupervised concepts) is discussed in appendix A.1", it is not discussed (or I can't find it). Would the authors mind adding this section?
* What intervention policy (i.e. choice of concept to intervene on) is currently used for Fig. 4? Also, I was thinking that maybe an additional benefit of having an LLM classifier would be that the LLM could decide on which concept to intervene on next, i.e. query the user based on information that would help it make a better decision.
* I did not understand the explanation of what D-GT means in Table 3. Can the authors try to explain it in more detail? Thanks!

---

> ### Author Response · Authors · 2025-11-22
> **Respectfully disagree with "mainly adapting LLM terminology to CBM terminology."**
>
> Thank you for the detailed critique and the recognition of our presentation and extensive experiments. However, we respectfully disagree with the assessment that our work is merely "adapting LLM terminology" of "prompt engineering". For the following reasons, we firmly believe that Chat-CBM represents a significant improvement to the CBM field and sincerely ask you to take the time to consider our perspective:
>
> 1. **Addressing Core CBM Limitations**
>    Chat-CBM's research motivation is rooted in solving a clear and pressing problem within the CBM field: the limitations of existing score-based classifiers. These limitations include the classifier's inability to reason about concept semantics; an intervention method restricted to modifying a fixed set of predicted scores; and the inability to effectively intervene on unsupervised CBMs. Our work explores a new approach to reasoning using a language-based classifier, which leverages the inherent concept semantics from the bottleneck layer itself. This is an exploratory effort directly tackling a core problem in CBMs, not an application of LLMs to a new domain.
>
> 2. **Comprehensive Experimental Validation**
>    We have provided sufficient experimental evidence to validate our method's soundness and effectiveness, particularly concerning model intervention performance. Beyond the inherent advantages of Chat-CBM's language-based interventions, we have demonstrated effective interventions on both supervised and unsupervised CBMs. This is particularly significant given that traditional unsupervised CBMs, with their large number of noisy and densely activated concepts, have been difficult to intervene upon. Furthermore, we demonstrated the ability to introduce new conceptual information at test time, under incomplete concept sets, intervention via high-level strategy guidance (appendix B.1), and a user-study (appendix B.2).
>
> 3. **More User-friendly Interpretability**
>    While we respect the option that an LLM is a black-box component without the parameter transparency of a linear layer, its autoregressive nature, which generates an explanation before a final prediction, ensures a form of transparency in the machine-user interaction. Our multi-faceted intervention experiments (e.g., correcting the input to a more accurate form and seeing a corresponding increase in accuracy) also validate the preservation of causality and faithfulness. And the complementary user experiments (appendix B.2) validate that Chat-CBM provides more intuitive and efficient model interaction and model control to end-users.
>
>
> We understand that the shift from a score-based classifier to a language-based classifier challenges the perceptions of much existing work. However, we believe that the reasons above, combined with the interactive, intervenable, and performance advantages offered by Chat-CBM, justify its exploration.
>
> ---
>
> Below, we respond to your concerns using updated results from our revision, which includes new non-linear baselines (MLPs), expanded intervention experiments, and clarified research focus with experiment support. We have incorporated these changes in the revision PDF (changes are marked with a blue font).

---

> ### Author Response · Authors · 2025-11-22
>
> ## W1. On Novelty: "Prompt Engineering" vs. Semantic Bottleneck
>
> We respectfully argue that Chat-CBM represents a comprehensive exploration of the feasibility and distinct advantages of shifting from a score-based to a language-based classifier. While our method leverages frozen LLMs, a neat design choice that you agree in strengths, this implementation detail should not obscure the fundamental paradigm shift we propose and explore.
>
> By transforming the bottleneck from a fixed numerical vector to a semantic space, we unlock intervention capabilities that are structurally impossible for standard CBMs. We have made the following improvements in revision:
>
> ### **(1) Clarifying Scope and Contribution**
>
> We rephase our claim in line 59-61 into "*these drawbacks of limited types of interventions and the ineffective intervention performance in some cases primarily stem from the reliance on score-based label predictors*", **to clearly state our focus on the intervention capabilities instead of simply improve performance by enlarging parameters**.
>
> ### **(2) Comprehensive Validation of Different Types of Interventions Chat-CBMs Support**
>
> To demonstrate that this is not just "chatbot conversation" but a rigorous functional upgrade, we conduct and expanded our experiments for the four types of interventions, covering all interaction modalities Chat-CBM enables:
>
> **1) Standard Numerical Intervention**: Same with the original version but rephrase the caption of Figure 4 into "Intervention via Standard Numerical Intervention".
>
> **2) Intervention via Concept Correction or Removal**: According to your suggestion that the assistant shouldn't know the ground truth class label, we conduct our experiment on the unsupervised datasets in a more rigorous manner with results updated in Figure 5 (Intervention via Simulated Concept Correction or Removal).
>
> Specifically, we now employ an assistant multimodal large language model (MLLM) to simulate user interventions. We provide the input image, the analysis and prediction generated by the Chat-CBM, the concept semantics utilized by the Chat-CBM (**which are the information a user can directly access**), and ask the MLLM to select only one action and one concept from <emphasize, concept> or <remove, concept>. And the output of the MLLM is converted into a **conversational intervention in natural language** such as "< concept > was wrongly predicted and doesn't exist in the image. Ignore this concept and answer again".  (Full details on the simulated procedure are provided in the appendix A.7).
>
> **Chat-CBM (V2C-CBM)** ( the results for Chat-CBM (LaBo) are also provided in Figure 5 and Table 12)
>
> | Dataset / N. | 0     | 1     | 2     | 3     | 4     | 5     |
> | ------------ | ----- | ----- | ----- | ----- | ----- | ----- |
> | CIFAR10      | 0.955 | 0.967 | 0.982 | 0.991 | 0.996 | 0.999 |
> | CIFAR100     | 0.727 | 0.803 | 0.853 | 0.891 | 0.925 | 0.943 |
> | Food-101     | 0.786 | 0.873 | 0.924 | 0.947 | 0.958 | 0.976 |
> | Flower-102   | 0.914 | 0.966 | 0.986 | 0.992 | 0.996 | 0.999 |
> | DTD          | 0.734 | 0.842 | 0.906 | 0.928 | 0.941 | 0.961 |
>
> The updated results in Figure 5 still support our statement that we can intervene with effective language-based concept correction/removal.
>
> **3) Concept Augmentation**: The experiment results in Figure 6 in the original version already demonstrate the feasibility and effectiveness of this type of intervention.
>
> To mitigate possible misunderstanding caused by the original caption "Intervention using new concepts", we rephase the paragraph caption and Figure 6's caption into "Intervention via Concept Augmentation".
>
> **4) High-level Strategy Guidance as Intervention**: Thank you for the suggestion, we originally provide an example case in Figure 3 (case 3) to show this kind of intervention. In the revision, we add an experiment on the SkinCon dataset (skin disease) by using the ABCDE skin assessment rule as the high-level strategy guidance as intervention, and show quantitative results for improving the accuracy via this kind of intervention.
>
> | Concept Accuracy=0.883 | CBM   | Chat-CBM (LLaMA3-8B) | Chat-CBM + ABCDE |
> | ---------------------- | ----- | -------------------- | ---------------- |
> | Class Accuracy         | 0.671 | 0.701                | 0.790            |
>
> Full details are provided in appendix B.1.
>
> (ABCDE stands for "Asymmetrical, Border, Color, Diameter, Evolving" which is an established assessment rule in clinical literature)
>
> ### (3) Honest Discussion of Trade-offs
>
> In the revision, we have added baseline with non-linear heads in Table 1 and Figure 4, and also Chat-CBM with 0.5B LLM in Table 5 for reference.
>
> And we have also expanded the discussion on limitations (Appendix D), acknowledging that while the frozen LLM enables these powerful interventions, it introduces high parameter counts and currently lacks awareness of concept uncertainty. And also provide failure case visualization and discussion in Appendix C.

---

> ### Author Response · Authors · 2025-11-22
>
> ## W2. Interpretability
>
> We thank you for the detailed feedback regarding your thought about the interpretability of language-based classifier. We understand your stance against equating LLM reasoning traces with the transparency of fully linear models.
>
> In response, we wish to state that, although the language-based label predictor doesn't have the perfect transparency (clear contribution for each concept to target), Chat-CBM still maintain the human-understandable concept-based interpretability, which is the core essential of CBMs:
>
> 1. Preservation of Concept-Based Fidelity: we deliberately preserve the concept bottleneck layer architecture. The prediction is still grounded in the semantic concept bottleneck layer, ensuring that the model's decision is concept-based.
> 2. Constrained Reasoning: analyzing the concept semantics to the label is a simpler task than complex, unconstrained CoT reasoning, as illustrated in the "analysis" part in Figure 3 and many other visualization results in the appendix. And we think that while the LLM’s internal mechanism is opaque, its output ('analysis' in Figure 3) provides a semantic justification that might be more intuitive and actionable for a human user than a list of numerical weights from a linear layer.
> 3. Empirical Validation of Fidelity: most importantly, the success of our intervention experiments empirically validates the fidelity of the LLM to the concept inputs. As concept quality is improved (via user intervention), Chat-CBM's prediction performance consistently improves, which shows the LLM is using the concept semantics for prediction, rather than relying on spurious prototype matching or opaque reasoning.
>
> **And we also conduct a user study** in appendix B.2, which actually validate that the language-based inference, explanation, and intervention provide more intuitive and efficient user experience.
>
> | Metric   | Intervention Time | Success Rate | User Satisfaction |
> | -------- | ----------------- | ------------ | ----------------- |
> | CBM      | 215.36 seconds    | 0.63         | 0.74              |
> | Chat-CBM | 100.62 seconds    | 0.82         | 0.84              |
>
> (Details of the user study are provided in Appendix B.2)

---

> ### Author Response · Authors · 2025-11-22
>
> ## W3. Conclusions drawn from the results
>
> ### (i) Non-linear Baselines
>
> We add new baselines with non-linear heads to Table 1 and Figure 4 (CBM-MLP(1) and CBM-MLP(2)), and Chat-CBM still outperforms or matches these non-linear baselines (e.g., on CUB: Chat-CBM 81.5% vs. CBM-MLP 79.1% in Table 1).
>
> Honestly, we have tried to build baseline with larger non-linear trainable heads but failed due to serve over-fitting, so we choose explicitly state that we use a 70B LLM in Table 1 and also provide other sizes in Table 5 in section 4.1:
>
> "*Table 1 presents the classification accuracy of different methods on datasets with concept labels. Chat-CBM with frozen LLaMA3-70B-Instruct surpasses the baselines on the CUB and AwA2 datasets, and Table 5 also exhibits results with different LLM sizes and series*"
>
> ### (ii) Few-shot Performance
>
> We want to highlight that Table 2 already provides both the all-shot and the few-shot (1-shot and 2-shot) performance of all baseline models for comparison. However, we concede that focusing the discussion in Section 4.2 only on the few-shot comparison is misleading. So we rewrite the paragraph to clearly state that :
>
> > "*Chat-CBM can achieve comparable performance under few-shot configurations, but is inferior to LaBo and V2C-CBM under the all-shot setting, though it's equipped with numerous parameters*"
>
> and explicitly point out that:
>
> > "*It may need additional effort to design more appropriate concept extraction methods for achieving competitive performance on unsupervised datasets with language-based label predictors, but currently, we choose to focus on exploring the intervention capabilities of Chat-CBM on unsupervised datasets.*"
>
> ### (iii) Intervention on Unsupervised Datasets
>
> We thank you for the suggestion that such simulated experiments should keep the real world setting that the user only can see the image and the output from the model without knowing the ground truth label.
>
> We conduct our experiment on the unsupervised datasets in a more rigorous manner with results updated in Figure 5 (Intervention via Simulated Concept Correction or Removal).
>
> Specifically, we now employ an assistant multimodal large language model (MLLM) to simulate user interventions. We provide the input image, the analysis and prediction generated by the Chat-CBM, the concept semantics utilized by the Chat-CBM (**which are the information a user can directly access**), and ask the MLLM to select only one action and one concept from <emphasize, concept> or <remove, concept>. And the output of the MLLM is converted into a **conversational intervention in natural language** such as "< concept > was wrongly predicted and doesn't exist in the image. Ignore this concept and answer again".  (Full details on the simulated procedure are provided in the appendix A.7).
>
> **Chat-CBM (V2C-CBM)** ( the results for Chat-CBM (LaBo) are also provided in Figure 5 and Table 12)
>
> | Dataset / N. | 0     | 1     | 2     | 3     | 4     | 5     |
> | ------------ | ----- | ----- | ----- | ----- | ----- | ----- |
> | CIFAR10      | 0.955 | 0.967 | 0.982 | 0.991 | 0.996 | 0.999 |
> | CIFAR100     | 0.727 | 0.803 | 0.853 | 0.891 | 0.925 | 0.943 |
> | Food-101     | 0.786 | 0.873 | 0.924 | 0.947 | 0.958 | 0.976 |
> | Flower-102   | 0.914 | 0.966 | 0.986 | 0.992 | 0.996 | 0.999 |
> | DTD          | 0.734 | 0.842 | 0.906 | 0.928 | 0.941 | 0.961 |
>
> The updated results in Figure 5 still support our statement that we can intervene with effective language-based concept correction/removal.
>
> ### (iv) Wikipedia experiment
>
> This experiment is to further demonstrate the ability to leverage new concept at test-time in supplement to the controlled incomplete concept settings. We rephrase our expression from "using Wikipedia description as intervention" to:
>
> "*We also explore the descriptions of the target class from Wikipedia and replace the class names with general names like "the bird" or "the animal", **and then provide them to Chat-CBMs as new information at test-time**. Although the huge performance improvement, as shown in Figure 6 (+wiki), seems obvious because of the rich information from Wikipedia, we want to argue that previous activation-based classifiers don't  easily support **receiving such kind of new information at test-time**},*"

---

> ### Author Response · Authors · 2025-11-22
>
> ## W4. Unsupervised Baselines on Noisy Datasets
>
> We would like to clarify our architectural focus again: our primary goal was to explore the functional benefits of transitioning from a score-based to a language-based classifier, and thus we intentionally preserved the original concept prediction step without modification in both supervised and unsupervised settings to isolate the differences between score-based and language-based classifiers.
>
> And our intervention results successfully demonstrate the advantage of Chat-CBM, we have shown significant improvements in intervention performance over unsupervised CBM baselines (Figure 5). Furthermore, in response to reviewer suggestions, we have further optimized the intervention setup for unsupervised datasets as shown in section 4.2 (B) and Figure 5.
>
> ## W5 & Q1. Reproducibility
>
> We will open-source our code after internal review if the paper got acceptance. And we believe that the architecture, built on standard CBM frameworks and publicly available frozen LLMs (which you agree that it is neat), is already highly reproducible, as stated in our reproducibility statement.
>
> ## W6. Awareness about Concept Uncertainty and Absence
>
> We agree with this point and add it as method limitation in the section 5, provide a detailed discussion of this limitation in appendix D, and also provide a failure case of Chat-CBM caused by this limitation in Figure 15.
>
> ## W7. The Flexibility to Add New Concepts at Test-time
>
> We respectfully disagree with the assessment that [3] targets the problem of adding new concepts at test time.
>
> The core distinction lies in the mechanism of intervention:
>
> 1.  **[3] (Laguna et al.)** focuses on enabling instance-specific interventions in **black-box models** by improving the consistency of intermediate representations ($z$) with a **fixed, pre-defined set of concepts** ($C$). Their intervention is a **numerical optimization** ($z \to z'$) designed to align the black-box activation vector with edited values ($c'$) that still belong to the original concept space $C$. This technique relies on a fixed probe ($q_\xi$) learned on annotated concepts and does not offer a mechanism for the model to understand or utilize new concepts that were not part of the initial concept set $C$.
>
> 2.  Our **Chat-CBM** achieves the flexibility to **add new concepts** and **external knowledge** by replacing the rigid, score-based classifier with a language-based classifier (LLM) that reasons over concept semantics, allowing the intervention interface to move beyond fixed numerical editing to incorporate high-level, natural-language guidance, including new concepts. This capability is fundamentally different from and substantially extends the scope of activation-editing methods like [3].
>
> Therefore, we think the flexibility offered by Chat-CBM is a novel advancement over prior works that were confined to editing or selecting from a fixed concept set. And also, Chat-CBM supports such kind of intervention via natural language, which is also shown to be effective in our small-scale user study in Appendix B.2
>
> ## W8. Test Set Leakage?
>
> Our experiments are restricted on the image classification benchmarks, while the LLM pretraining is conducted on text corpus. The two data source have no overlap so we think there should not be test set leakage problem. And the LLM is frozen and used strictly for inference based on the provide concept semantics (predicted by the concept predictor) without any finetuning.
>
> And the general knowledge of LLM is the reason why we can use frozen LLM and also enable flexible conversational intervention. So we do not think there are test data leakage problem regarding the task accuracy on these image classification benchmarks.

---

> ### Author Response · Authors · 2025-11-22
>
> ## Other Questions
>
> **Q2: SCBM**: we have added the results of SCBM in Table 1 and Figure 4.
>
> **Q3: Discussion on Top-N choice of concepts**: we have provided this discussion in appendix A.4 in the revision:
>
> "*Both LaBo and V2C-CBM utilize a large number of concepts for prediction (typically 50 $\times$ N\_class). When these concepts are ranked by activation values, usually only the top 10 to 15 concepts are highly relevant to the image's semantics. Because Chat-CBM reasons on texts, the concepts with confusing semantics can have a noticeable negative impact. During development, we experimented with settings like top-5, 10, 15, 20 on small experiments. We found that the top-10's performance is the most robust. Therefore, this hyperparameter depends on the chosen concept extractor's performance, but might be determined by manually inspecting how many top concepts are generally relevant to the input image.*"
>
> **Q4: Intervention policy in Figure 4**: we use random policy in Figure 4, and we have added this information in section 4.2.
>
> **Q5: D-GT and D**: We apologize for the confusion.  $\boldsymbol{D}$-GT refers to the standard ICL format used in our main experiments, where the demonstration prompt contains only the concept semantics and the correct label (without baseline class candidates). The strategy labeled simply as  $\boldsymbol{D}$ (without the checkmark) is an ablation variant where we explicitly *added* the baseline's class candidates ($Y_{\rm val}$) into the prompt. We found that the cleaner $\boldsymbol{D}$-GT format yields better results and adopted it as the default.
>
> We have update our explanation in section 4.3 for clarity.
>
> ---
>
> In conclusion, With the new baselines (MLP), user study, re-designed intervention experiments on unsupervised datasets, and more thorough experiment for evaluating each new intervention method that Chat-CBM enables, we hope to have demonstrated that Chat-CBM offers a tangible improvement in the trade-off between accuracy, interpretability, and interactivity.
>
> And we hope these revisions and clarifications can address some of your concerns and sincerely request that you can reexamine the value of Chat-CBMs and reconsider your score.

---

### Official Review · Reviewer_gDGJ · 2025-10-31

**Soundness:** 4
**Presentation:** 3
**Contribution:** 3
**Rating:** 6
**Confidence:** 4

**Summary:**

This paper introduces Chat-CBM, a novel approach to Concept Bottleneck Models that replaces traditional score-based classifiers with language-based classifiers using frozen LLMs. The key innovation lies in reasoning directly over concept semantics rather than numerical activations, which enables richer interventions including concept correction, addition/removal, and high-level reasoning guidance. The authors evaluate their approach on nine datasets, demonstrating improvements in both supervised and unsupervised CBM settings while maintaining interpretability.

**Strengths:**

1. Well-motivated problem: Clearly articulates limitations of existing CBMs (restricted interventions, struggles with unsupervised settings) and how the proposed approach addresses them.

2. Comprehensive evaluation: Nine datasets spanning supervised/unsupervised settings, multiple baseline comparisons, and extensive ablations demonstrate thoroughness.

3. Practical intervention capabilities: Figure 3 and Figure 5 convincingly show that conversational interventions work in practice and can substantially improve performance.

4. Honest limitations discussion: Appendix B provides an assessment of computational costs, incomplete concept challenges, and safety concerns with LLM knowledge.

5. Strong performance on key metrics: The intervention curves (Figure 4-5) show Chat-CBM can approach or exceed fully-supervised baselines with just a few interventions. Preview TeX is supported

**Weaknesses:**

1. The paper's central value proposition is enabling "richer and more intuitive interventions" through natural language, yet all experimental validation uses automated procedures without any human participants. Section 4.2's intervention experiments (Figure 5, lines 374-398) rely on an assistant LLM with ground-truth labels rather than actual users, making it impossible to verify whether real humans find these interactions genuinely intuitive or efficient. The authors should conduct even a small user study with 15-20 participants comparing Chat-CBM against traditional CBM interventions, measuring metrics like time-to-correct-prediction and user satisfaction to validate their core claims about interaction quality.

2. The paper experiments exclusively with large LLMs (8B-72B parameters in Table 5) without exploring whether compact models like 1-3B parameter LLMs could achieve acceptable performance with feasible resource requirements. The authors should add experiments establishing a performance-efficiency trade-off curve with smaller models and provide concrete deployment recommendations, or clearly acknowledge that the approach is currently limited to research settings rather than practical interactive systems.

3. The paper demonstrates Chat-CBM works when concept predictions are accurate (93-98% in Table 1) but provides no systematic analysis of robustness to concept errors or

minimum quality thresholds needed for effectiveness. The paper only shows cherry-picked success cases in Figures 3 and 7-11 without any failure analysis, leaving practitioners unable to assess when they can trust the model's reasoning. The authors should add experiments with artificially corrupted concepts to establish robustness boundaries, include a failure analysis subsection with concrete error examples, and provide guidelines like minimum concept accuracy thresholds so users understand the approach's reliability limits.

4. The experimental design gives Chat-CBM advantages not available to baselines, most notably using validation set statistics to construct class priors θ in unsupervised settings (Section A.4, lines 802-829), which is only revealed in the appendix rather than transparently discussed in the main paper. Additionally, the paper mentions XBM and CB-LLM as conceptually similar work using language models for interpretable classification (Section 2.2, lines 126-137) but provides no empirical comparison. The authors should include direct comparisons with XBM and CB-LLM on several datasets, add ablations where baselines also receive class prior knowledge to ensure fair comparison, and explicitly state in the main paper what information each method accesses.

**Questions:**

1. Concept selection: How sensitive is performance to the choice of top-N concepts (you use top-10 for unsupervised)? Have you experimented with learned concept selection based on uncertainty or relevance?

2. Figure 5 shows sequential interventions, but does the model maintain coherent reasoning across turns? Can conflicting interventions cause degradation?

3. How does Chat-CBM handle strongly correlated concepts? Do LLMs naturally account for concept dependencies, or does this require explicit modeling?

4. What happens when users introduce concepts at test time that are semantically distant from training concepts? Is there a notion of "valid" interventions?

---

> ### Author Response · Authors · 2025-11-22
>
> Thank you for the detailed review and the encouraging assessment. We have updated the paper to address your concerns, including a user study (appendix B.2), failure case analysis (appendix C), small LLM, and clarification on research focus and experimental fairness.
>
> ## W1. Human User Study
>
> Firstly, we follow CBM, CEM, and ECBM to validate interactivity by measuring performance gains against the number of interventions, a metric we also report . And we have shown improved intervention efficiency in Fig 4, 5 and 6.
>
> And to empirically validate the intuitive nature of Chat-CBM following your suggestion, we develop a interactive interface during the rebuttal period, and conduct a user study on the CUB dataset. The details of the experiment setting, the user interface are provided in Appendix B.2, and we explicitly note this experiment in section 4.2.
>
> The user study validates that the language-based inference, explanation, and intervention provide more intuitive and efficient user experience.
>
> | Metric   | Intervention Time | Success Rate | User Satisfaction |
> | -------- | ----------------- | ------------ | ----------------- |
> | CBM      | 215.36 seconds    | 0.63         | 0.74              |
> | Chat-CBM | 100.62 seconds    | 0.82         | 0.84              |
>
> ## W2. LLM Size
>
> We have expanded our analysis to include compact models to Table 5 and add a discussion in section 4.3.
>
> - **Trade-off:** As expected, performance scales with model size. Using **Qwen2.5-0.5B-Instruct**, classification accuracy on CUB decreases (0.725 vs. 0.801 with 14B), but the model remains functional.
> - **Deployment:** We identify two viable paths for practitioners: (1) Cloud deployment using large frozen LLMs via APIs for maximum reasoning capability, or (2) Edge deployment by fine-tuning compact models (like the 0.5B variant) on task-specific data. Our reported results with frozen small models serve as a performance lower bound.
>
> ## W3. Robustness and Failure Analysis
>
> **Failure cases analysis**: We have added a dedicated section (Appendix C.2) with visualizations of failure cases (Figures 14 & 15). Failures primarily stem from: (1) Noisy concept predictions that mislead the LLM (e.g., predicting "longneck" for a deer), and (2) Lack of uncertainty awareness in the language predictor (e.g., difficulty distinguishing Cow vs. Ox when visual concepts are identical).
>
> **Robustness to concept errors**:
>
> 1. Sensitivity: Because Chat-CBM relies on semantic reasoning, it can be sensitive to wrong concepts if they have strong semantic implications.
> 2. Mitigation: However, we demonstrate two forms of robustness:
>    1. Incomplete Concepts: Figure 6, Tables 16 & 17 show Chat-CBM outperforms independent CBMs even when concept supervision is drastically reduced (e.g., 0.620 vs. 0.591 on CUB with only 56 concepts), proving the LLM can reason with partial information. This means we can use less accurate concept for mitigating the influence of concept errors.
>    2. Class Priors: As shown in Table 3, incorporating the class prior significantly mitigates the impact of concept noise (improving accuracy from 0.645 to 0.771). (we discuss the fairness of incorporating class prior in the next question)

---

> ### Author Response · Authors · 2025-11-22
>
> ## W4. experimental design & other baselines
>
> **Fairness of the Experimental Design**
>
> First, we clarify that we use **training set statistics**, not validation data,, to construct the class priors, as detailed in Appendix A.5. This is now explicitly stated in **Section 3.2.2** and detailed in **Appendix A.5**.
>
> The only information derived from the validation set is the selection of In-Context Learning (ICL) examples. Since baselines also use the validation set for model selection (i.e., selecting the best-performing checkpoint) prior to testing, we believe this comparison remains fair. Furthermore, the ability to leverage ICL examples is an inherent architectural advantage of using language-based predictors, which cannot be directly applied to standard numerical classifiers.
>
> **Other Baselines**
>
> **(1) XBMs** employ multimodal models (such as BLIP-2 or LLaVA) to generate image captions that serve as concepts, subsequently training a BERT model on these captions. The primary motivation behind XBMs is to leverage MLLMs to mitigate the annotation costs associated with building CBMs, rather than focusing on user intervention procedures.
>
> Notably, replacing generated captions with Ground Truth (GT) image captions in XBM does not yield a significant performance improvement on the CUB dataset. In contrast, replacing the concept predictions in Chat-CBM with GT concepts dramatically boosts classification accuracy from 0.797 to 0.998:
>
> | Model                                            | Class Accuracy |
> | ------------------------------------------------ | -------------- |
> | XBM-BLIP (BLIP-ViT-B32 + trainable BERT)         | 0.810          |
> | XBM-BLIP + Intervention (ration=1.0)             | 0.822          |
> | Chat-CBM (ResNet101 + frozen LLaMA3-8B-Instruct) | 0.797          |
> | Chat-CBM + Intervention (ratio=1.0)              | 0.998          |
>
> In this work, our primary focus is exploring the intervention capabilities enabled by a language-based label predictor. Therefore, we maintain a consistent concept predictor backbone (ResNet-101 or CLIP-ViT-L14) to isolate the contribution of our proposed method.
>
> **(2) CB-LLM** focuses specifically on text classification tasks. Crucially, CB-LLM still relies on a score-based label predictor, utilizing the LLM merely as the backbone for text representation. Consequently, we did not include CB-LLM in our direct comparisons.
>
> ## Q1. concept selection sensitivity
>
> The sensitivity of the performance to the choice of top-N concepts for unsupervised datasets, mainly depends on the concept quality of the concept extractor. We add a discussion of this in appendix A.4:
>
> "*Both LaBo and V2C-CBM utilize a large number of concepts for prediction (typically 50 $\times$ N\_class). When these concepts are ranked by activation values, usually only the top 10 to 15 concepts are highly relevant to the image's semantics. Because Chat-CBM reasons on texts, the concepts with confusing semantics can have a noticeable negative impact. During development, we experimented with settings like top-5, 10, 15, 20 on small experiments. We found that the top-10's performance is the most robust. Therefore, this hyperparameter depends on the chosen concept extractor's performance, but might be determined by manually inspecting how many top concepts are generally relevant to the input image.*"
>
> The concept extraction process of LaBo and V2C-CBM is actually measuring the CLIP-similarity between the image feature and the text feature, and the similarity score between 0-1 is used as the predicted concept probabilities. We think this is similar to select concepts based on relevance. In addition, the way of using top-activated concepts is similar to selecting concepts with low uncertainty (except we don't use the non-activated concepts)
>
> ## Q2. coherent reasoning & conflicting intervention
>
> Chat-CBM naturally maintains context via the conversation history ($\cal H$).
>
> In this work, we consider that the user interventions should be correct. But if a user provides a conflicting intervention (contradicting a previous turn), the model naturally prioritizes the latest instruction (recency bias) while attempting to maintain logical consistency. We believe this behavior is desirable as it remains faithful to the user's most recent intent.

---

> ### Author Response · Authors · 2025-11-22
>
> ## Q3. handle strongly correlated concepts
>
> Chat-CBM handles correlations implicitly via the frozen LLM's pre-trained world knowledge. For example, the model understands that "wings" and "fly" are semantically linked without requiring the explicit covariance modeling used in baselines like SCBM.
>
> ## Q4. concepts that are semantically distinct & notion of valid interventions
>
> As demonstrated in Figure 6, we successfully intervened using Wikipedia descriptions (e.g., habitat, sound) that are semantically distant from the training attributes. A "valid" intervention is simply any concept within the LLM's open-vocabulary understanding. This allows Chat-CBM to leverage multimodal or external knowledge that standard CBMs cannot process.
>
> ---
>
> We hope these clarifications, supported by our additional experiments can address your concerns. If you have any other questions, please let us know.

---

### Official Review · Reviewer_u7B8 · 2025-10-31

**Soundness:** 3
**Presentation:** 3
**Contribution:** 2
**Rating:** 6
**Confidence:** 4

**Summary:**

The authors propose Chat-CBM, a model that leverages a large language model (LLM) as the classifier in the prediction phase. Chat-CBM integrates concept semantics directly into the prediction process: labels are inferred through reasoning over the semantics of concepts rather than through activation scores. This design preserves the interpretable aspect of Concept Bottleneck Models (CBMs) — namely, the concept layer.

This modification also enables richer forms of intervention, since interventions can now be expressed directly in natural language within the LLM. In contrast, a simple linear layer only allows interventions on the numerical scores of each concept. Furthermore, the flexibility of the LLM makes it possible to incorporate prior knowledge about concept semantics.

The authors identify several types of possible interventions: traditional numerical interventions as in standard CBMs, but also natural language interventions that allow users to correct a concept, add or remove concepts, or provide higher-level information.

Experiments are conducted in both supervised settings (three datasets with concept labels) and unsupervised settings (six datasets without concept labels). Two LLMs are evaluated: LLaMA3-Instruct and Qwen2.5-Instruct.

**Strengths:**

- The idea of replacing the classification layer with an LLM is not highly original, but the experiments and comparisons with existing models are interesting and well executed.
- The authors explore several types of interventions that are not possible with standard CBMs, highlighting the added flexibility of their approach.
- Chat-CBM generally achieves higher accuracy than baseline models in the supervised setting across the three tested datasets, though this improvement is observed mainly in the 1-shot and 2-shot scenarios.
- Concept correction interventions lead to a faster improvement in accuracy compared to the other models evaluated.

**Weaknesses:**

- The question of whether the overall interpretability of the model is preserved after introducing an LLM is not addressed.
- The authors do not discuss the contribution of the LLM’s prior knowledge to the observed accuracy improvements, in comparison to the contribution of the concept layer itself.
- The use of different LLMs for the supervised and unsupervised settings is not clearly justified, although an ablation study is provided for part of the unsupervised experiments.

**Questions:**

The paper presents an interesting approach by replacing the classification layer of a CBM with an LLM, which enables new forms of intervention. However, several points would benefit from clarification or deeper discussion:

- The authors should mention explicitly that they limit their experiments to image inputs, even if this does not undermine the generality of the proposed approach. Is the method easily transferable to non-visual modalities?
- What is the relative contribution of the frozen LLM to the prediction accuracy, compared to that of the concept layer?
- To what extent is interpretability preserved at the LLM stage, especially if the LLM is not guaranteed to be faithful in its reasoning?
- In Table 1, what is the difference between the two rows labeled Chat-CBM?
- Why is LLaMA3 used in some experiments and Qwen2.5 in others? The motivation for this choice is not clearly explained, although an ablation study is provided for part of the unsupervised setting.

Minor Comments (no impact on the evaluation score):
- The best-performing model’s score is not always highlighted in bold in the tables.
- Line 776: there seems to be a character encoding issue.

---

> ### Author Response · Authors · 2025-11-22
>
> We thank the reviewer for the constructive feedback and the recognition of our work. We are encouraged by the positive assessment. Below, we address the specific questions and weaknesses raised for better clarity.
>
> ## Q1. Transferability to non-visual modalities.
>
> We add explicit statement at the last paragraph in the section 1 (introduction), the first paragraph in section 3.1  (method: problem definition and method overview) and section 4 (experiments: datasets) to state that our focus and experiments are limited to image classification tasks.
>
> We think the method can be transferable to non-visual modalities if proper concept extractor exists, because the proposed Chat-CBM is mainly consist of: (1) concept extraction, (2) knowledge-enhanced semantic concept bottleneck layer, and (3) a frozen LLM as the language-based label predictor, where (2) is built on (1), and (3) is frozen without special tuned or design. Another reason is that the frozen LLM have shown good generalization ability across many tasks these years.
>
> ## Q2 & W2. Contribution of LLM Prior vs. Concept Layer to the accuracy.
>
> Our results indicate that the **concept bottleneck layer remains the foundational driver of performance**, while the LLM acts as a reasoning engine that refines these inputs. We provide three lines of evidence:
>
> **Performance Drops with Incomplete Concepts (Figure 6 & Table 13):** In our new experiments, when we deliberately reduced the number of available concepts (e.g., training with only a subset), the performance dropped significantly. This proves the LLM is not "hallucinating" correct answers from its prior but relies heavily on the signal from the concept layer.
>
> **Ablation of Priors (Table 3)**: As shown in Table 3, when adding the class prior knowledge into the semantic concept bottleneck layer ($\boldsymbol{\theta}$), we see steady performance improvements across different benchmarks and LLMs.
>
> **Scaling Trends (Table 5):** Although larger LLMs provide better reasoning (improving accuracy), the performance gain is bounded by the concept quality, confirming the concept layer is the primary information bottleneck.
>
> ## Q3 & W1. Preservation of Interpretability
>
> Although the language-based label predictor doesn't have the perfect transparency (clear contribution for each concept to target), we think that Chat-CBM still maintain the human-understandable concept-based interpretability, which is the core essential of CBMs. We validate this through:
>
> 1. Preservation of Concept-Based Fidelity: we deliberately preserve the concept bottleneck layer architecture. The prediction is still grounded in the semantic concept bottleneck layer, ensuring that the model's decision is concept-based.
> 2. Constrained Reasoning: analyzing the concept semantics to the label is a simpler task than complex, unconstrained CoT reasoning, as illustrated in the "analysis" part in Figure 3 and many other visualization results in the appendix. And we think that while the LLM’s internal mechanism is opaque, its output ('analysis' in Figure 3) provides a semantic justification that might be more intuitive and actionable for a human user than a list of numerical weights from a linear layer.
> 3. Empirical Validation of Fidelity: most importantly, the success of our intervention experiments empirically validates the fidelity of the LLM to the concept inputs. As concept quality is improved (via user intervention), Chat-CBM's prediction performance consistently improves, which shows the LLM is using the concept semantics for prediction, rather than relying on spurious prototype matching or opaque reasoning.
>
> And we also conduct a user study in appendix B.2, which actually validates that the language-based inference, explanation, and intervention provide more intuitive and efficient user experience.
>
> | Metric   | Intervention Time | Success Rate | User Satisfaction |
> | -------- | ----------------- | ------------ | ----------------- |
> | CBM      | 215.36 seconds    | 0.63         | 0.74              |
> | Chat-CBM | 100.62 seconds    | 0.82         | 0.84              |
>
> (Details of the user study are provided in Appendix B.2)

---

> ### Author Response · Authors · 2025-11-22
>
> ## Q4. Tow rows of Chat-CBM in Table 1
>
> This is to validate that Chat-CBM can be implemented with different concept extractors/CBM baselines.
>
> The first row is the result of Chat-CBM with CBM as the baseline, and the second row is Chat-CBM with ECBM as the baseline.
>
> ## Q5 & W3. The motivation of using different LLMs
>
> We mainly use LLaMA3-8B-Instruct for experiments on supervised datasets and we provide detailed ablation in Table 5 for supervised datasets.
>
> For unsupervised datasets which usually contain more samples, we find that the performance gain from further enlarging the LLM size from 32B is relatively small, so we choose to use the Qwen2.5-32B for the main experiments on unsupervised datasets.
>
> These results of the use of different LLM families and sizes across different scenarios demonstrates the robustness and generalizability of our method.
>
>
>
> ## Q6. Minor Comments
>
> Thank you for the detailed review and we have correct these issues in the revision version.
>
> ---
>
> We hope these clarifications, supported by our additional experiments can address your concerns. If you have any other questions, please let us know.

---

### Official Review · Reviewer_LX7g · 2025-11-01

**Soundness:** 3
**Presentation:** 2
**Contribution:** 2
**Rating:** 4
**Confidence:** 4

**Summary:**

The paper proposes Chat-CBM, an interactive concept bottleneck framework that replaces the standard score-based label head with a language-based classifier over concept semantics using a frozen LLM. The idea is to preserve CBM interpretability (predictions pass through a concept set) while enabling richer interventions: concept correction, adding/removing concepts, injecting external knowledge, and high-level reasoning guidance. The method targets both supervised and unsupervised CBMs (where concept activations are noisy and dense). Experiments on nine datasets show accuracy gains over conventional CBMs, and interaction studies (including an assistant-LLM that performs automatic interventions) indicate that a few edits can surpass “all-shot” baselines. Ablations explore knowledge injection, ICL shot counts, and LLM size, and a “new concepts at test time” setting (including Wikipedia text) illustrates flexibility.

**Strengths:**

1 CBMs are interpretable but often brittle, and their linear heads constrain user intervention.
2 Experiments are comprehensive, including both supervised/unsupervised settings, plus ablations on (i) knowledge priors, (ii) ICL shots, (iii) LLM families/sizes.
3 The writing is easy to follow

**Weaknesses:**

1 The central interaction result relies on an assistant LLM that edits concepts. As such, the quality of concepts may be dependent on the LLMs used.
2 Comparisons focus on classical CBMs (linear heads, activation-based). Missing are stronger non-linear concept heads.
3 The paper is similar to the paper "Interpreting Pretrained Language Models via Concept Bottlenecks". Can the authors distinguish their work from this one?
4 The datasets used seem to be simple and not very complex, using CIFAR and ImageNet.

**Questions:**

How is this paper related to the paper I mentioned in the weakness? Can the work be generalized to more complex tasks like QA?

---

> ### Author Response · Authors · 2025-11-22
>
> We sincerely thank you for recognizing the importance of our problem setting, the comprehensiveness of our experiments, and the clarity of our writing. We appreciate your constructive feedback, which has helped us significantly strengthen the paper.
>
> Below, we address your concerns point-by-point, highlighting new experiments and clarifications added to the revised manuscript (changes are marked with blue font in the revision PDF).
>
> ## W1. more rigorous intervention experiments on unsupervised datasets
>
> The intervention experiments on the standard numerical intervention (Figure 4) and via adding new concepts (Figure 6) are conducted by intervening using ground truth concept labels on the supervised datasets, instead of leveraging assistant LLMs.
>
> For results on the unsupervised dataset, **we conduct our experiment on the unsupervised datasets in a more rigorous manner with results updated in Figure 5 (Intervention via Simulated Concept Correction or Removal).**
>
> Specifically, we now employ an assistant multimodal large language model (MLLM) to simulate user interventions. We provide the input image, the analysis and prediction generated by the Chat-CBM, the concept semantics utilized by the Chat-CBM (**which are the information a user can directly access**), and ask the MLLM to select only one action and one concept from <emphasize, concept> or <remove, concept>. And the output of the MLLM is converted into a **conversational intervention in natural language** such as "< concept > was wrongly predicted and doesn't exist in the image. Ignore this concept and answer again".  (Full details on the simulated procedure are provided in the appendix A.7).
>
> **Chat-CBM (V2C-CBM)** ( the results for Chat-CBM (LaBo) are also provided in Figure 5 and Table 12)
>
> | Dataset / N. | 0     | 1     | 2     | 3     | 4     | 5     |
> | ------------ | ----- | ----- | ----- | ----- | ----- | ----- |
> | CIFAR10      | 0.955 | 0.967 | 0.982 | 0.991 | 0.996 | 0.999 |
> | CIFAR100     | 0.727 | 0.803 | 0.853 | 0.891 | 0.925 | 0.943 |
> | Food-101     | 0.786 | 0.873 | 0.924 | 0.947 | 0.958 | 0.976 |
> | Flower-102   | 0.914 | 0.966 | 0.986 | 0.992 | 0.996 | 0.999 |
> | DTD          | 0.734 | 0.842 | 0.906 | 0.928 | 0.941 | 0.961 |
>
> The updated results in Figure 5 still support our statement that we can intervene with effective language-based concept correction/removal.
>
> **And we also conduct a small-scale user study** in appendix B.2, which actually validate that the language-based inference, explanation, and intervention provide more intuitive and efficient user experience.
>
> | Metric   | Intervention Time | Success Rate | User Satisfaction |
> | -------- | ----------------- | ------------ | ----------------- |
> | CBM      | 215.36 seconds    | 0.63         | 0.74              |
> | Chat-CBM | 100.62 seconds    | 0.82         | 0.84              |
>
> (Details of the user study are provided in Appendix B.2)
>
> ## W2. non-linear baselines
>
> We add new baselines with non-linear heads to Table 1 and Figure 4 (CBM-MLP(1) and CBM-MLP(2)), and Chat-CBM still outperforms or matches these non-linear baselines (e.g., on CUB: Chat-CBM 81.5% vs. CBM-MLP 79.1% in Table 1).
>
> We have also tried to build baseline with larger non-linear trainable heads (e.g., larger MLP or transformer haeds) but failed due to serve over-fitting, which is likely attributable to the inherent challenge of training large score-based classifiers directly on numerical vectors of concept activations.
>
> So in the revision, we choose explicitly state that we use a 70B LLM in Table 1 and also provide other sizes in Table 5 in section 4.1:
>
> "*Table 1 presents the classification accuracy of different methods on datasets with concept labels. Chat-CBM with frozen LLaMA3-70B-Instruct surpasses the baselines on the CUB and AwA2 datasets, and Table 5 also exhibits results with different LLM sizes and series*"

---

> ### Author Response · Authors · 2025-11-22
>
> ## W3 & Q1. Relationship to "Interpreting Pretrained Language Models via Concept Bottlenecks"
>
> We appreciate the reviewer's attention to related work. And we have also added this paper into section 2.2 as our related work.
>
> After reading this paper, we find that both studies involve concept bottleneck models and pretrained (large) language models. However, we assert that our proposed method, Chat-CBM, introduces a distinct architecture and paradigm shift for CBM inference and intervention, addressing the crucial limitations of prior score-based label predictors, including the CBE-PLMs (concept bottleneck-enabled pretrained language models) and its training framework, C$^3$M.
>
> **So the core distinction lies in the architecture of the label predictor and the resulting intervention capabilities**. And we provide more detailed discussion about the difference between the two works.
>
> 1. **Score-based vs. language-based classifier**:
>
>    1. **CBE-PLM/C$^3$M framework**: uses a score-based classifier on the activations of the concept bottleneck layer. (" For NLP tasks such as sentiment analysis, concepts can be Food, Am biance, and Service as shown in Figure 1, where each concept corresponds to a neuron in the con cept bottleneck layer. The final decision layer is then a linear function of these concepts")
>    2. **Chat-CBM**: replaces the score-based classifier with language-based classifier (frozen LLMs in our paper) that reasons directly in the semantic space over the extract concept semantics
>
>    This is a key architectural divergence: CBE-PLM learns a linear mapping from concept scores $\hat{c} \rightarrow y$, whereas Chat-CBM performs classification from concept semantics $\hat{s} \rightarrow y$.
>
> 2. **Enhanced Intervention Capabilities**:
>
>    The architectural difference directly governs the type of user intervention supported, which is the primary focus of Chat-CBM:
>
>    1. **CBE-PLM/C$^3$M framework**: interventions are restricted to the **numerical edits of the predicted concept activations**. ("We follow Koh et al. (2020) to intervene in the predicted concepts and investigate the impact of such interventions on test-time prediction accuracy")
>    2. **Chat-CBM**: in addition to standard concept activation correction (Fig.2), **Chat-CBM also supports richer intervention types that are hard to implement for score-based classifier.** These include: **(1)** language-based concept correction (Fig.3 case 1); **(2)** concept augmentation/removal (Fig.3 case 2; Fig.6); **(3)** high-level strategy guidance (Fig.3 case 3). And also exhibits effective interventions for fully-unsupervised (without any expert-defined concepts) CBMs (Fig.5).
>
> 3. **Distinct Roles of LLMs**:
>
>    Both works use LLMs and pretrained language models, but for different purposes:
>
>    1. **CBE-PLM/C$^3$M framework:** LLMs (ChatGPT) are primarily used as a data augmentation tool to generate concepts and noisy pseudo-labels for the training phase (Sec 4.1). The LLM is not the label predictor at test time. The goal is to train a robust score-based CBM.
>       And other pretrained language models (GPT2, BERT, RoBERTa) serve as the feature extractor for concept prediction (analogous to the ResNet-101 for concept prediction in our work).
>    2. **Chat-CBM:** A frozen LLM (LLaMA3, Qwen2.5) is the label predictor during the inference phase (Sec 3.1, Fig 2). It performs semantic reasoning based on the extracted concepts and user input. The focus is on interactive performance and semantic grounding at test time, rather than training robustness.
>
> In conclusion, the CBE-PLMs/C$^3$M paper focuses on robustly training a conventional score-based CBM for interpretable text classification. Our Chat-CBM paper proposes a new CBM architecture with language-based label predictor which enables flexible, conversational interventions. This novel intervention capability is the primary contribution and distinction of our work.
>
> ## W4. using CIFAR and ImageNet
>
> We appreciate your review. In our paper, we conduct experiments across nine datasets including:
>
> 1. Fine-grained classification: CUB (birds), AwA2 (animals), Flower-102, Food-101. And they are standard benchmarks for CBMs.
> 2. Specialized domains: DTD (texture classification), PBC (a medical dataset for white blood cell classification), we also add a new dataset SkinCon in appendix B.1 in revision for evaluating the capabilities of high-level strategy guidance as intervention.
> 3. General: ImageNet, CIFAR 10/100
>
> In addition, Table 2 in the original paper already presents Chat-CBM's performance on CIFAR-10/100 and ImageNet, and Figure 5 shows intervention experiments on CIFAR-10/100. We believe this sufficiently demonstrates Chat-CBM's effectiveness on the datasets you mentioned.

---

> ### Author Response · Authors · 2025-11-22
>
> ## Q2. generalized to more complex tasks like QA?
>
> We are very appreciative of the insightful and forward-thinking extension proposed by the reviewer.
>
> Our current work, Chat-CBM, primarily focuses on introducing a novel CBM architecture that employs a language-based label predictor to enable richer, conversational interactive concept intervention capabilities, which we have validated mainly on image classification tasks and compared with existing CBM methods.
>
> Regarding the generalization of this strategy to QA tasks, we believe its feasibility hinges on task-specific concept definition and extraction mechanisms.
>
> 1. Extensibility to QA: Extractive or generative QA tasks often involve locating or producing an answer $A$ based on a question $Q$ and an input context $C$. If it is possible to reasonably define a set of general, interpretable concepts (e.g., entity types, relational logic, temporal order) for a given class of QA tasks, and to design a concept extractor with sufficient generalizability, then the core philosophy of Chat-CBM could potentially be extended to the QA domain.
> 2. Potential Benefit: The introduction of a concept bottleneck layer may enable ante-hoc interpretability in QA, allowing users to inspect and intervene in the conceptual reasoning process before the model outputs the final answer.
>
> While we value this helpful extension, we believe this promising avenue is best reserved for future exploration.
>
> ---
>
> We sincerely hope that you can raise your score if the above responses address your concerns. If your have any other questions, please let us know.

---

### Official Review · Reviewer_DF9V · 2025-11-03

**Soundness:** 3
**Presentation:** 3
**Contribution:** 2
**Rating:** 2
**Confidence:** 4

**Summary:**

The paper proposes to integrate LLMs into the CBM architecture. The LLM serves to augment the mapping from concepts to labels through reasoning over concept semantics. Chat-CBM enhances the concept bottleneck layer by providing additional information such as the top-N most likely class given the concept activation, learning examples, and concept priors for relevant classes. Moreover, Chat-CBM allows a novel way to intervene at test time—through conversation with the LLM. End users can correct, add, or remove concepts and provide higher-level strategy guidance. The authors run experiments on both supervised and unsupervised concept datasets.

**Strengths:**

- Novel and interesting idea
- Variety of benchmarks in experimental section
- Figures are mostly well designed and help readers

**Weaknesses:**

I don't think directly comparing existing CBM architecture with a linear concept bottleneck layer to Chat-CBM is fair. Chat-CBM is augmented with a significantly more capable model (with billions of parameters). As a result, I am not convinced that the authors have shown that
>  these challenges primarily stem from the reliance on score-based label predictors (line 59-61).

What it's really showing is that a linear classifier mapping concepts to labels is not as performant/expressive as a larger model; this is a pretty trivial result.

To me, Chat-CBM's value (and novelty) is in test-time intervention -- the ability to intervene with natural language. In addition to concept correction (analogous to numerical intervention in traditional CBMs), the authors state that we can intervene with Concept augmentation/removal or High-level strategy guidance. However, Section 4.2 does not investigate the effects (or gains) of these intervention strategies that is unique to Chat-CBM.

Furthermore, there are a few unsubtantiated claims throughout this paper. For example:
> This process allows users to understand and take control of the decision pipeline (line 298-299).

> The introduced prior knowledge effectively enhances Chat-CBM’s robustness against noise in concept predictions (line 430-431).

In both cases, I am left asking "how exactly?"

Overall, it would help if the authors (1) elaborate on findings/results (i.e., what significance they have) and (2) back claims with evidence.

**Questions:**

- How would Chat-CBM perform on tasks where there is little public knowledge? I imagine LLMs have a substantial amount of text/corpus from the tasks in the benchmarks.
- How exactly do you intervene on datasets without concept labels? There is no "true" label here.
> employ an assistant LLM that selects concepts to emphasize, remove, or augment based on the conversation history of Chat-CBM, the top-20 predicted concepts, and the ground truth class label.

How does one know the ground truth class label at test time?

- Is $\hat{y} = \textrm{arg} \max_y P(y | D, \theta, \hat{s})$ correct? The LLM does not output numerical probabilities...
- Do the authors think that natural language interventions are more "costly" for the end user at test time? It seems like one may have to put in more thought into formulating their intervention.

---

> ### Author Response · Authors · 2025-11-22
>
> We sincerely thank the reviewer for the detailed feedback and for acknowledging the novelty of our idea. We understand your concerns regarding the fairness of the comparison, the scope of intervention experiments, and the need for empirical evidence to back our claims.
>
> In our revision, we have significantly expanded the experiments and analysis to address these points, and also update our manuscript as shown in the rebuttal revision PDF file (changes are marked with a blue font).
>
> ## W1. Fairness of Comparison
>
> **More baselines**: We add new baselines with non-linear heads to Table 1 and Figure 4 (CBM-MLP(1) and CBM-MLP(2)), and Chat-CBM still outperforms or matches these non-linear baselines (e.g., on CUB: Chat-CBM 81.5% vs. CBM-MLP 79.1% in Table 1).
>
> **Clarifying our claim and focus**: We rephase our claim in line 59-61 into " these drawbacks of limited types of interventions and the ineffective intervention performance in some cases primarily stem from the reliance on score-based label predictors", **to clearly state our focus on the intervention capabilities** instead of simply improve performance by enlarging parameters.
>
> ## W2. Evaluation of Unique Intervention Methods
>
> In the section 4.2 of the revision manuscript, we experiment with different types of intervention methods to validate the interactivity and intervention capabilities of Chat-CBMs, including:
>
> **1) Standard Numerical Intervention**: Same with the original version but rephrase the caption of Figure 4 into "Intervention via Standard Numerical Intervention".
>
> **2) Intervention via Concept Correction or Removal**: We conduct our experiment on the unsupervised datasets in a more rigorous manner with results updated in Figure 5 (Intervention via Simulated Concept Correction or Removal).
>
> Specifically, we now employ an assistant multimodal large language model (MLLM) to simulate user interventions. We provide the input image, the analysis and prediction generated by the Chat-CBM, the concept semantics utilized by the Chat-CBM (**which are the information a user can directly access**), and ask the MLLM to select only one action and one concept from <emphasize, concept> or <remove, concept>. And the output of the MLLM is converted into a **conversational intervention in natural language** such as "< concept > was wrongly predicted and doesn't exist in the image. Ignore this concept and answer again".  (Full details on the simulated procedure are provided in the appendix A.7).
>
> **Chat-CBM (V2C-CBM)** ( the results for Chat-CBM (LaBo) are also provided in Figure 5 and Table 12)
>
> | Dataset / N. | 0     | 1     | 2     | 3     | 4     | 5     |
> | ------------ | ----- | ----- | ----- | ----- | ----- | ----- |
> | CIFAR10      | 0.955 | 0.967 | 0.982 | 0.991 | 0.996 | 0.999 |
> | CIFAR100     | 0.727 | 0.803 | 0.853 | 0.891 | 0.925 | 0.943 |
> | Food-101     | 0.786 | 0.873 | 0.924 | 0.947 | 0.958 | 0.976 |
> | Flower-102   | 0.914 | 0.966 | 0.986 | 0.992 | 0.996 | 0.999 |
> | DTD          | 0.734 | 0.842 | 0.906 | 0.928 | 0.941 | 0.961 |
>
> The updated results in Figure 5 still support our statement that we can intervene with effective language-based concept correction/removal.
>
> **3) Concept Augmentation**: The experiment results in Figure 6 in the original version already demonstrate the feasibility and effectiveness of this type of intervention.
>
> To mitigate possible misunderstanding caused by the original caption "Intervention using new concepts", we rephrase the paragraph caption and Figure 6's caption into "Intervention via Concept Augmentation".
>
> **4) High-level Strategy Guidance as Intervention**: Thank you for the suggestion, we originally provide an example case in Figure 3 (case 3) to show this kind of intervention. In the revision, we add an experiment on the SkinCon dataset (skin disease) by using the ABCDE skin assessment rule as the high-level strategy guidance as intervention, and show quantitative results for improving the accuracy via this kind of intervention.
>
> | Concept Accuracy=0.883 | CBM   | Chat-CBM (LLaMA3-8B) | Chat-CBM + ABCDE |
> | ---------------------- | ----- | -------------------- | ---------------- |
> | Class Accuracy         | 0.671 | 0.701                | 0.790            |
>
> Full details are provided in appendix B.1.
>
> (ABCDE stands for "Asymmetrical, Border, Color, Diameter, Evolving" which is an established assessment rule in clinical literature)

---

> ### Author Response · Authors · 2025-11-22
>
> ## W3. Substantiating Claims
>
> **1) Users take control**: We additionally conduct a user study (appendix B.2) to validate that users have more intuitive and effective control of the model behavior. Results in Table 15 show that users achieved a higher success rate (Chat-CBM 82% vs 63% CBM) and report higher satisfaction with Chat-CBM compared to standard CBMs. And the user interface in Figure 8 may also provide evidence that the language-based intervention procedure can facilitate users to understand and take control the decision.
>
> **2) prior knowledge enhances robustness**: the ablation study in Table 3 shows that adding prior knowledge ($\boldsymbol{\theta}$) can improve the classification accuracy of Chat-CBMs (e.g., CUB accuracy improves from 0.645 to 0.771). This empirical evidence supports our claim that priors help stabilize predictions against noisy concept inputs.
>
> In the revision, we directly add an example after this claim which points to Table 3 ("The introduced prior knowledge effectively enhances Chat-CBM's robustness against noise in concept predictions (e.g., adding $\boldsymbol{\theta}$ improves the accuracy on the CUB dataset from 0.645 to 0.771 as shown in Table 3)")
>
> ## Q1. Performance on Tasks with Little Public Knowledge
>
> This is an insightful question. We acknowledge that since we utilize frozen LLMs, we further conduct experiment on the  the **SkinCon** dataset (skin disease) with Chat-CBM (frozen LLaMA3-8B-Instruct), which has less knowledge about this special domain (**Appendix B.1**).
>
> | Concept Accuracy=0.883 | CBM   | Chat-CBM (LLaMA3-8B) |
> | ---------------------- | ----- | -------------------- |
> | Class Accuracy         | 0.671 | 0.701                |
>
> Despite being a specialized dermatology task, Chat-CBM successfully utilized the provided concept definitions and diagnostic criteria to achieve better performance than baseline CBMs, especially after intervention via the ABCDE skin assessment rule. This suggests that as long as the domain knowledge can be described textually (e.g., via prompts or definitions), Chat-CBM generalizes well without gradient updates.
>
> ## Q2. Intervention on Datasets without Concept Labels
>
> We have updated results with our newly designed experiments, as explained in above in "2. Evaluation of Unique Intervention Methods", we prompt a MLLM with only the input image and the predictions from Chat-CBM to assess the validity of visual concepts, without any hints on the true class label. Full details are provided in section 4.2 and the appendix A.7 in the revision PDF.
>
> And in the new experiments, none of the model knows the ground truth class label which is more aligned with the real world setting.
>
> ## Q3. LLM *does* Outputs Numerical Probabilities
>
> Although the final output of LLMs is text, the auto-regressive generation process of an LLM fundamentally involves calculating and utilizing numerical probabilities (by selecting the argmax token for greedy decoding, or by sampling from the resulting probability distribution). **Therefore, the LLM *does* output numerical probabilities at its core.**
>
> Furthermore, similar mathematical representations are very common in the LLM literature, such as the Eq 1-2 in "A Survey on In-context Learning (EMNLP 2024)." for your reference.
>
> ## Q4. Are natural language interventions more "costly" for the user?
>
> On the contrary, we think that natural language interventions are actually less costly for the end users. And this thought is also validated by our study as detailed in appendix B.2.
>
> We find that intervention time for Chat-CBM was significantly lower (average time 100.62 seconds for Chat-CBMs but 215.36 seconds for standard CBMs.)
>
> The main reason for the above results is that in standard CBMs with 100+ concepts, users struggle to *find* the right slider to adjust (cognitive load of search). In Chat-CBM, users simply type what they see (e.g., "The bill is actually red"), which is faster and more intuitive.
>
> ---
>
> Overall, we have further improved our manuscript with clarified claims and new evidence (experiments), and have incorporated these changes into our manuscript as shown in the revision PDF.
>
> We hope these revisions and clarifications address your concerns and demonstrate the distinct value of Chat-CBM and sincerely request that you can reconsider your score. If you have any other questions, please let us know.

---

### Meta-Review · Area_Chair_vQ6j · 2026-01-07

**Summary:**

The proposal replaces the linear classifier in CBMs with a language-based classifier that reasons directly over the concepts.  The proposal provides additional information such as top-N most likely classes given the concept activation, learning examples, and concept priors of the relevant class.

The reviewers highlighted the unfair comparisons, the lack of evaluations with other LLM-based methods, and distinction against them.  Moreover, the use of an opaque method, like an LLM, defeats the purpose of using a concept layer that is supposed to shed light and provide interpretability into the models.  The paper received mixed reviews, with a trend towards reject.  Given the main issues about the lack of interpretability and unfair comparisons, I recommend the rejection of the paper.

Strengths:
- Interesting idea to mix LLMs and CBMs
- Varied benchmarks in the experiments
- Explored several types of interventions

Weaknesses:
- Unfair comparisons between linear layers and complex LLMs
- Lack of investigation of the effects of the intervention strategies
- The central interaction results relies on an assistant LLM that edits the concepts
- Lack of comparison and distinction against similar works that use LLMs
- It is unclear whether the interpretability is preserved after using an LLM
- The impact of the LLM prior knowledge in the improvements is unclear
- The experiments use different LLMs for the supervised and unsupervised cases without justification

**Reviewer Concerns:**

Reviewer DF9V raised concerns about unfair comparisons, unsubstantiated claims, lack of elaboration on the results, and unclear method to intervene on datasets without labels.  The authors replied and claimed that the focus of the work was on the interventions instead of the performance, and expanded on the use of the LLMs to process the concepts.

Reviewer LX7g raised concerns about the reliance on LLMs as an assistant to edit the concepts, and the lack of comparisons against stronger non-linear concept heads, as well as distinguishing themselves against prior work that relies on language models.  The authors provided additional results and explanations to the reviewer questions.  However, the issues about unfair comparisons and missing LLM-based methods remained open.

Reviewer u7B8 raised concerns about the overall impact of the LLMs in the CBM interpretability, the prior knowledge in the LLM contribution to the final accuracy, and the different LLMs for the different settings. The authors addressed most of the concerns.  However, there are still open issues w.r.t. the fairness and validity of the comparisons between the proposal and other LLM-based methods.

Reviewer gDGJ raised concerns about the evaluation since the paper claims intuitive interventions but the experiments are not done with humans, the lack of comparisons against smaller models, and the lack of evaluation to concept errors.  The authors replied to the reviewer's concerns.  While most of them were addressed, there were still open issues with small enough LLMs to be comparable to the linear layers, and no clear experiments that demonstrate the interpretability of the proposal.

Reviewer 8iQG raised concerns about the use of LLMs and in-context learning for a similar task, the lack of interpretability by using an opaque LLM instead of the interpretable linear layer, and the unfair comparisons.  The authors replied to the questions and concerns of the reviewer.  Despite the replies, I shared the concerns from the reviewer about the lack of interpretability and unfair comparisons in the paper.

**Reviewer Scores:**

Reviewer DF9V recommended a rejection.  While the authors replied to the reviewer's concerns, the main issue about the fairness and the comparisons against CBMs remained.

Reviewer LX7g recommended a weak reject.  The authors' rebuttal doesn't address the issues with unfair comparisons and against stronger baselines to differentiate the gains from the CBM and the LLMs.

Reviewer u7B8 recommended a weak accept. However, the remaining issues about the fairness of the comparisons remain an open issue that lowers the score.

Reviewer gDGJ recommended a weak accept.

Reviewer 8iQG recommended a reject.  Despite the authors' reply, the main issues about the fair comparisons and lack of interpretability remain open.

---

### Decision · Program_Chairs · 2026-01-26

Reject